# Ether lipid biosynthesis promotes lifespan extension and enables diverse pro-longevity paradigms in *Caenorhabditis elegans*

Lucydalila Cedillo[1,2,3†], Fasih M Ahsan[1,2,3†], Sainan Li[1,2], Nicole L Stuhr[4], Yifei Zhou[1,2], Yuyao Zhang[1,2], Adebanjo Adedoja[1,2,3], Luke M Murphy[1,2,3], Armen Yerevanian[1,2], Sinclair Emans[1,2], Khoi Dao[5], Zhaozhi Li[6], Nicholas D Peterson[7], Jeramie Watrous[5], Mohit Jain[5], Sudeshna Das[6], Read Pukkila-Worley[7], Sean P Curran[4], Alexander A Soukas[1,2]*

[1]Center for Genomic Medicine and Diabetes Unit, Endocrine Division, Department of Medicine, Massachusetts General Hospital and Harvard Medical School, Boston, United States; [2]Broad Institute of Harvard and MIT, Cambridge, United States; [3]Program in Biological and Biomedical Sciences, Division of Medical Sciences, Harvard Medical School, Boston, United States; [4]Leonard Davis School of Gerontology, University of Southern California, Los Angeles, United States; [5]Department of Medicine and Pharmacology, University of California San Diego, San Diego, United States; [6]Biomedical Informatics Core, Massachusetts General Hospital and Harvard Medical Schoo, Cambridge, United States; [7]Program in Innate Immunity, Division of Infectious Diseases and Immunology, University of Massachusetts Medical School, Worcester, United States

*For correspondence:
asoukas@mgh.harvard.edu

†These authors contributed equally to this work

Competing interest: The authors declare that no competing interests exist.

**Abstract** Biguanides, including the world's most prescribed drug for type 2 diabetes, metformin, not only lower blood sugar, but also promote longevity in preclinical models. Epidemiologic studies in humans parallel these findings, indicating favorable effects of metformin on longevity and on reducing the incidence and morbidity associated with aging-related diseases. Despite this promise, the full spectrum of molecular effectors responsible for these health benefits remains elusive. Through unbiased screening in *Caenorhabditis elegans*, we uncovered a role for genes necessary for ether lipid biosynthesis in the favorable effects of biguanides. We demonstrate that biguanides prompt lifespan extension by stimulating ether lipid biogenesis. Loss of the ether lipid biosynthetic machinery also mitigates lifespan extension attributable to dietary restriction, target of rapamycin (TOR) inhibition, and mitochondrial electron transport chain inhibition. A possible mechanistic explanation for this finding is that ether lipids are required for activation of longevity-promoting, metabolic stress defenses downstream of the conserved transcription factor *skn-1*/Nrf. In alignment with these findings, overexpression of a single, key, ether lipid biosynthetic enzyme, *fard-1*/FAR1, is sufficient to promote lifespan extension. These findings illuminate the ether lipid biosynthetic machinery as a novel therapeutic target to promote healthy aging.

## Editor's evaluation

This paper explores the molecular basis underlying metformin treatment to understand why it is such an effective drug for improving age-related health and lifespan. Using *C. elegans* as a model organism in which to do this, the paper hones in on the role of ether lipid biosynthesis as an effector

of metformin, and more broadly as a process implicated in extending lifespan in response to diet, TOR signalling and mitochondrial based interventions. The compelling data substantially support the conclusions and the better understanding of biguanide impact on metabolism is highly important in the field.

## Introduction

Metformin is the first line therapy for type 2 diabetes and the most frequently prescribed oral hypoglycemic medication worldwide (*Inzucchi et al., 2012*). Human epidemiologic studies note an association between metformin use and decreased incidence of cancer (*Evans et al., 2005*; *Yuan et al., 2013*). In addition, metformin extends lifespan in invertebrate and vertebrate models (*Cabreiro et al., 2013*; *Martin-Montalvo et al., 2013*; *Onken and Driscoll, 2010*), and therefore may reduce aging-related diseases in humans (*Barzilai et al., 2016*). Nonetheless, our understanding of the molecular pathways governing the health-promoting effects of metformin is only just beginning to emerge. Our previous work identified a conserved signaling axis connecting mitochondria, the nuclear pore complex, and mTORC1 inhibition that is required for metformin-mediated extension of lifespan in *Caenorhabditis elegans* and inhibition of growth in worms and human cancer cells (*Wu et al., 2016*). The energy sensor AMP-activated protein kinase (AMPK) is not necessary for metformin-induced growth inhibition in *C. elegans* but is required for the drug's pro-longevity effects (*Cabreiro et al., 2013*; *Onken and Driscoll, 2010*; *Chen et al., 2017*). Consistently, mechanistic studies indicate that the longevity-promoting transcription factor SKN-1/nuclear factor erythroid 2-related factor (Nrf) is required for biguanide-mediated lifespan extension (*Cabreiro et al., 2013*; *Onken and Driscoll, 2010*). The relationship of these metformin longevity response elements to each other and their hierarchy in the biological response to biguanides remains unknown. Thus, the mechanisms by which metformin exacts its beneficial effects on health are likely to be branching and complex.

The importance of ether lipids, a major structural component of cell membranes, to aging and longevity is not fully established. Ether lipids are involved in the maintenance of general membrane fluidity and in the formation of lipid rafts within microdomains, which are important for promotion of membrane fusion and cellular signaling (*Glaser and Gross, 1994*; *Komljenovic et al., 2009*; *Marrink and Mark, 2004*). Ether lipids have broad roles in the regulation of cell differentiation (*Davies et al., 2001*; *Facciotti et al., 2012*; *Rodemer et al., 2003*; *Teigler et al., 2009*), cellular signaling (*Thukkani et al., 2002*; *Albert et al., 2003*), and reduction of oxidative stress through their action as antioxidants (*Morand et al., 1988*; *Zoeller et al., 1988*; *Reiss et al., 1997*; *Maeba et al., 2002*). Humans deficient in ether lipid biogenesis suffer from rhizomelic chondrodysplasia punctata (RCDP), a rare genetic disorder, which results in skeletal and facial abnormalities, psychomotor retardation, and is uniformly fatal typically before patients reach their teenage years (*White et al., 2003*). Thus, current evidence linking alterations in ether lipid levels to aging and longevity in humans is strictly correlative (*Gonzalez-Covarrubias et al., 2013*; *Pradas et al., 2019*).

Ether lipids, which are structurally distinct from canonical phospholipids, have a unique biosynthetic pathway through which a fatty alcohol is conjugated to the glycerol backbone at the *sn-1* position via an ether linkage. Ether lipid precursors are first synthesized by enzymes associated with the membranes of peroxisomes (*Ghosh and Hajra, 1986*; *Hardeman and van den Bosch, 1989*; *Singh et al., 1993*.) The main enzymes involved in ether lipid biosynthesis within the peroxisomal matrix are glyceronephosphate *O*-acyltransferase (GNPAT) and alkylglycerone phosphate synthase (AGPS). Fatty acyl-CoA reductase 1 (FAR1) supplies most of the fatty alcohols used to generate the ether linkage in the precursor, 1-*O*-alkyl-glycerol-3-phosphate. This precursor is then trafficked to the endoplasmic reticulum (ER) for acyl chain remodeling to produce various ether lipid products (*Hua et al., 2017*). In *C. elegans,* loss-of-function mutations of any of the three main enzymes involved in human ether lipid biosynthesis, *acl-7*/GNPAT, *ads-1*/AGPS, and *fard-1*/FAR1, result in an inability to produce ether-linked lipids, as in humans, and has been reported to shorten worm lifespan (*Drechsler et al., 2016*; *Shi et al., 2016*). Worms and human cells deficient in ether lipids exhibit compensatory changes in phospholipid species, including increases in phosphatidylethanolamines and phosphatidylcholines containing saturated fatty acids (*Rodemer et al., 2003*; *Benjamin et al., 2013*). However, in contrast to humans, ether lipid deficient nematodes develop to adulthood at a normal rate, providing an

**eLife digest** Metformin is the drug most prescribed to treat type 2 diabetes around the world and has been in clinical use since 1950. The drug belongs to a family of compounds known as biguanides which reduce blood sugar, making them an effective treatment against type 2 diabetes.

More recently, biguanides have been found to have other health benefits, including limiting the growth of various cancer cells and improving the lifespan and long-term health of several model organisms. Epidemiologic studies also suggest that metformin may increase the lifespan of humans and reduce the incidence of age-related illnesses such as cardiovascular disease, cancer and dementia. Given the safety and effectiveness of metformin, understanding how it exerts these desirable effects may allow scientists to discover new mechanisms to promote healthy aging.

The roundworm *Caenorhabditis elegans* is an ideal organism for studying the lifespan-extending effects of metformin. It has an average lifespan of two weeks, a genome that is relatively easy to manipulate, and a transparent body that enables scientists to observe cellular and molecular events in living worms.

To discover the genes that enable metformin's lifespan-extending properties, Cedillo, Ahsan et al. systematically switched off the expression of about 1,000 genes involved in *C. elegans* metabolism. They then screened for genes which impaired the action of biguanides when inactivated. This ultimately led to the identification of a set of genes involved in promoting a longer lifespan. Cedillo, Ahsan et al. then evaluated how these genes impacted other well-described pathways involved in longevity and stress responses.

The analysis indicated that a biguanide drug called phenformin (which is similar to metformin) increases the synthesis of ether lipids, a class of fats that are critical components of cellular membranes. Indeed, genetically mutating the three major enzymes required for ether lipid production stopped the biguanide from extending the worms' lifespans. Critically, inactivating these genes also prevented lifespan extension through other known strategies, such as dietary restriction and inhibiting the cellular organelle responsible for producing energy. Cedillo, Ahsan et al. also showed that increasing ether lipid production alters the activity of a well-known longevity and stress response factor called SKN-1, and this change alone is enough to extend the lifespan of worms.

These findings suggest that promoting the production of ether lipids could lead to healthier aging. However, further studies, including clinical trials, will be required to determine whether this is a viable approach to promote longevity and health in humans.

---

opportunity to determine the biological roles of ether lipids in aging and longevity without pleiotropies associated with developmental rate.

Here, we show that the ether lipid biosynthetic machinery is necessary for lifespan extension stimulated by metformin or the related biguanide phenformin in *C. elegans*. Metabolomic analysis indicates that phenformin treatment drives increases in multiple phosphatidylethanolamine-containing ether lipids through direct biguanide action on *C. elegans* rather than on the bacterial food source. Interestingly, requirement for the ether lipid biosynthetic genes extends to multiple genetic longevity paradigms, including defective mitochondrial electron transport function (*isp-1*), defective pharyngeal pumping/caloric restriction (*eat-2*), and compromises in mTOR complex 1 activation (*raga-1*). We show that overexpressing *fard-1*, the enzyme that produces fatty alcohols for ether lipid biogenesis in *C. elegans*, extends lifespan, supportive of the idea that alterations in the ether lipid landscape alone is sufficient to promote healthy aging. Mechanistically, ether lipids promote longevity downstream of biguanide action through activation of metabolic stress defenses and somatic lipid redistribution driven by the transcription factor SKN-1/Nrf. These data suggest that a heretofore unappreciated role for ether lipids is to enable organismal-level, longevity-promoting stress defenses.

# Results

## Genes responsible for ether lipid biosynthesis are necessary for biguanide-induced lifespan extension

A prior screen of ~1000 metabolic genes for RNA interference (RNAi) knockdowns that interfere with the growth-inhibitory properties of a high, 160 mM dose of metformin in *C. elegans* (utilized to maximize the sensitivity and specificity of our assay to identify true epistatic candidates) (*Wu et al., 2016*), yielded *fard-1* and *acl-7*, which are required for ether lipid biosynthesis. Ether lipids are distinguished from canonical phospholipids as the latter contain exclusively fatty acids conjugated to glycerol, whereas ether lipids contain a fatty alcohol conjugated to the glycerol backbone at the *sn-1* position via an ether linkage (*Figure 1A*). Confirming our screen results, granular, quantitative analysis following RNAi knockdown of *fard-1* and *acl-7* reveals significant resistance to biguanide-induced growth inhibition (*Figure 1—figure supplement 1A*). Our lab has previously demonstrated that biguanide effects on growth in *C. elegans* share significant overlap mechanistically with the machinery by which metformin extends lifespan in the worm, thus suggesting that modulation of ether lipid biosynthesis may also be responsible for the lifespan-extending properties of the drug (*Wu et al., 2016*). Indeed, loss-of-function mutations in any of three genes encoding enzymes required for ether lipid biosynthesis, *fard-1*, *acl-7*, or *ads-1*, significantly abrogate lifespan extension induced by lifespan-extending doses of metformin (50 mM) and the related biguanide phenformin (4.5 mM) (*Figure 1B–G*). Loss-of-function of *ads-1* and *acl-7* may display a modest increase in lifespan with metformin administration but display a percentage median lifespan increase significantly reduced in comparison to wild-type controls (*Figure 1B–G*, and throughout manuscript see *Supplementary file 1* for all tabular survival statistics and biological replicates). Confirming that these mutations confer resistance to metformin by compromising ether lipid synthetic capacity, RNAi knockdowns of *fard-1* and *acl-7* in wild-type worms also partially impair lifespan extension promoted by phenformin (*Figure 1—figure supplement 1B–C*). This dependency is not confounded by chemical inhibition of reproduction, as lifespan analyses performed without the use of the thymidylate synthase inhibitor 5-fluoro-2′-deoxyuridine (FUdR) reveal similar abrogation of biguanide-mediated lifespan extension with inactivation of the ether lipid synthetic machinery (*Figure 1—figure supplement 2A–F*; *Van Raamsdonk and Hekimi, 2011*). Studies from this point forward are presented predominantly with phenformin because phenformin is more readily absorbed without need for a specific transporter, unlike metformin (*Wu et al., 2016*; *Sogame et al., 2009*; *Segal et al., 2011*), and our experience indicates more consistent lifespan extension with phenformin in *C. elegans*.

Because ether lipids are a major structural component of cell membranes, one possibility is that deficiencies in ether lipid synthesis compromises drug action by reducing biguanide bioavailability in the worm. To test this, we compared the relative levels of biguanides present in vehicle- and biguanide-treated wild-type animals to the three ether lipid synthesis mutants by liquid chromatography-tandem mass spectrometry (LC-MS/MS). A comparison of normalized concentrations of phenformin across all four strains shows that phenformin abundance is quantitatively similar across wild-type and the three ether lipid mutant strains (*Figure 1H* and *Figure 1—figure supplement 1D*). Similar results were obtained when comparing levels of metformin in wild-type vs. ether lipid mutant animals (*Figure 1I* and *Figure 1—figure supplement 1E*). Thus, a deficiency in ether lipid synthesis does not significantly impact levels of biguanides in metformin- and phenformin-treated *C. elegans*.

## Phenformin induces changes in ether lipid levels

We reasoned that if biguanides require ether lipid biosynthesis to promote lifespan extension, that phenformin may promote synthesis of one or more ether lipids. To investigate the impact of biguanides on ether lipids at a high level, we first utilized gas chromatography-mass spectrometry (GC-MS) analysis. We first recapitulated the observation that *fard-1* mutants show absence of 18-carbon containing fatty alcohol derivatives (dimethylacetals [DMAs], which indicate alkenyl ether lipid or plasmalogen levels) and an accumulation of stearate (18:0) relative to wild-type controls by GC-MS (*Figure 2A—B*; *Shi et al., 2016*). We then asked if phenformin impacts the levels of 18-carbon alkenyl ether lipids in wild-type animals and if those corresponding changes are absent in *fard-1* mutants. Strikingly, phenformin-treated wild-type worms display a significant increase in 18:0 DMA versus vehicle, whereas no such increase is evident in drug-treated *fard-1* worms (*Figure 2C*). In addition, relative proportions of stearic acid (18:0) levels within the total fatty acid pool are significantly increased in

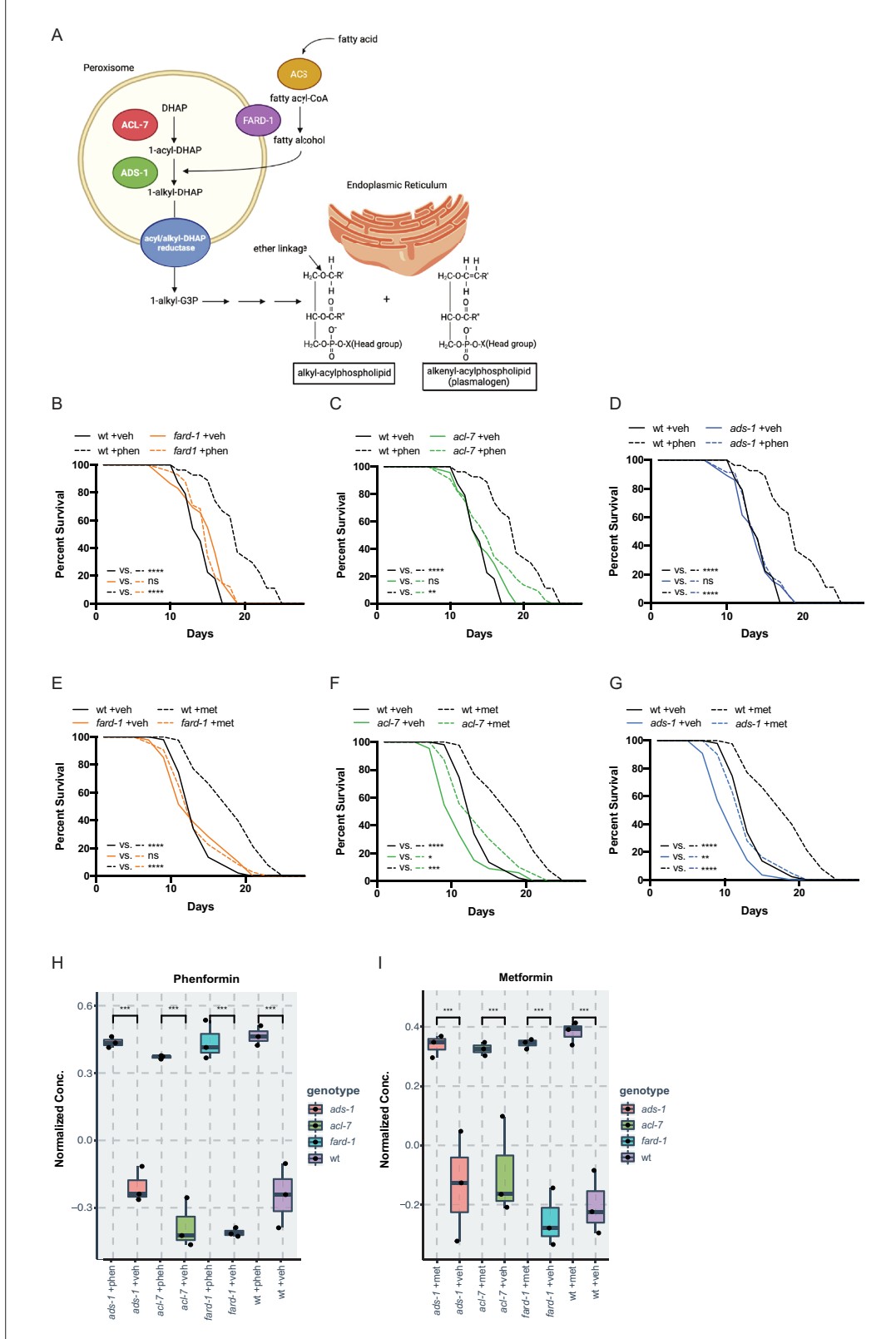

**Figure 1.** Genes responsible for ether lipid biosynthesis are necessary for biguanide-induced lifespan extension. (**A**) *C. elegans* ether lipid synthesis is catalyzed by three enzymes: fatty acyl reductase FARD-1, acyltransferase ACL-7, and alkylglycerone phosphate synthase ADS-1 (adapted from Figure 1 of ***Shi et al., 2016*** and ***Dean and Lodhi, 2018***). The latter two are localized to the peroxisomal lumen. (**B–D**) Missense, loss-of-function

*Figure 1 continued on next page*

*Figure 1 continued*

mutations in *fard-1* (**B**), *acl-7* (**C**), and *ads-1* (**D**) in *C. elegans* suppress phenformin-induced lifespan extension. (**E–G**) A deficiency of ether lipid synthesis in *fard-1* (**E**), *acl-7* (**F**), and *ads-1* (**G**) worm mutants blunts metformin-induced lifespan extension. Results are representative of three biological replicates. *, p<0.05; **, p<0.01; ***, p<0.001; ****, p<0.0001 by log-rank analysis. Note that (B–D) and (E–G) contain the same wild-type (wt) controls as they are visualized from the same replicate of the study. See also *Figure 1—figure supplement 1* and refer to *Supplementary file 1* for tabular survival data and biological replicates. (**H–I**) Normalized concentrations of phenformin (**H**) and metformin (**I**) in vehicle, 4.5 mM phenformin, or 50 mM metformin-treated wt *C. elegans* versus *fard-1*, *acl-7*, and *ads-1* mutants. n=3 biological replicates; ***, p<0.004 by two-tailed Student's t-test with Bonferroni correction for multiple hypothesis testing. Box represents 75th/25th percentiles, while whisker represents higher/lower hinge ± [1.5 * interquartile range (IQR)].

The online version of this article includes the following figure supplement(s) for figure 1:

**Figure supplement 1.** Reduced function of genes responsible for ether lipid biosynthesis partially suppresses biguanide effects of growth and lifespan without affecting biguanide levels.

**Figure supplement 2.** The use of 5-fluoro-2'-deoxyuridine (FUdR) in lifespan analyses does not impact the observed epistases between the ether lipid machinery and biguanide-mediated lifespan extension.

---

*fard-1* mutants treated with phenformin versus vehicle-treated *fard-1* controls (*Figure 2D*). In comparison, the relative proportion of stearic acid does not rise in phenformin-treated wild-type animals, suggesting that stearate is being utilized for ether lipid production. Analysis of the total fatty acid pool by GC-MS (*Figure 2—figure supplement 1*) indicates that aside from several fatty acids (e.g. 18:2), the most pronounced differences were in the plasmalogen pool. In alignment, an assessment of levels of additional alkenyl fatty alcohols in phenformin-treated, wild-type animals indicates a parallel, significant increase in the less abundant 16:0 DMA and 18:1 DMA species (*Figure 2E*). We conclude that phenformin treatment leads to an overall increase of alkenyl ether lipid levels in *C. elegans*.

To investigate relative changes in individual ether lipid abundance in response to phenformin at high resolution, we utilized LC-MS/MS analysis. Using this method, we detected 20 alkyl and alkenyl phosphatidylethanolamine-based ether lipids previously noted to be the most abundant ether lipids in *C. elegans* (*Drechsler et al., 2016*; *Shi et al., 2016*; *Figure 2F–G* and *Figure 2—source data 1*). This analysis indicates that phenformin treatment results in a significant increase in normalized abundance of four ether lipids, PE(O-16:0/18:1), PE(O-18:0/18:3), PE(O-18:0/20:2), and PE(P-18:1/18:1), even when corrected for multiple hypothesis testing. Most ether lipids measured display mean levels that increase with phenformin treatment, though these changes are either nominally significant or exhibit a nonsignificant trend because of the strict threshold required to reach significance when correcting for multiple hypotheses. Finally, phosphatidylethanolamine ether lipid abundances were extremely low in *fard-1*, *acl-7*, and *ads-1* mutants and unchanged by phenformin treatment, unlike in wild-type animals (*Figure 2F* and *Figure 2—source data 1*). In aggregate, these data indicate that phenformin treatment leads to increased abundance of multiple ether lipid species in *C. elegans*.

## Peroxisomal ether lipid synthesis is essential to the biological action of phenformin

In order to begin to understand the governance of ether lipid biosynthesis by biguanides, we examined the expression of a *C. elegans* FARD-1::RFP translational reporter, under the control of its own promoter (*Figure 2—figure supplement 2A*). Exogenously expressed FARD-1 *(fard-1 oe1)* is expressed in the intestine and localizes near structures resembling lipid droplets by Nomarski microscopy (*Figure 2—figure supplement 2B*). Given that ether lipid biogenesis occurs between peroxisomes and the ER (*Ghosh and Hajra, 1986*; *Hardeman and van den Bosch, 1989*; *Singh et al., 1993*; *Hua et al., 2017*), we crossed this FARD-1::RFP reporter to an animal bearing a GFP reporter that illuminates peroxisomes in the intestine (GFP fused to a C-terminal peroxisomal targeting sequence 1 [PTS1]) to determine if localization of FARD-1 is regulated by biguanides. FARD-1 does not possess a predicted PTS, in contrast to ACL-7 and ADS-1. At baseline, FARD-1::RFP fluorescence partially overlaps with peroxisomally targeted GFP (*Figure 2—figure supplement 2C*). Colocalization analysis indicates that treatment with phenformin does not change the amount of overlap between FARD-1::RFP and GFP::PTS1 relative to vehicle-treated controls (*Figure 2—figure supplement 2D*). To confirm our earlier observation that suggests FARD-1 colocalization with lipid droplets, we used confocal imaging

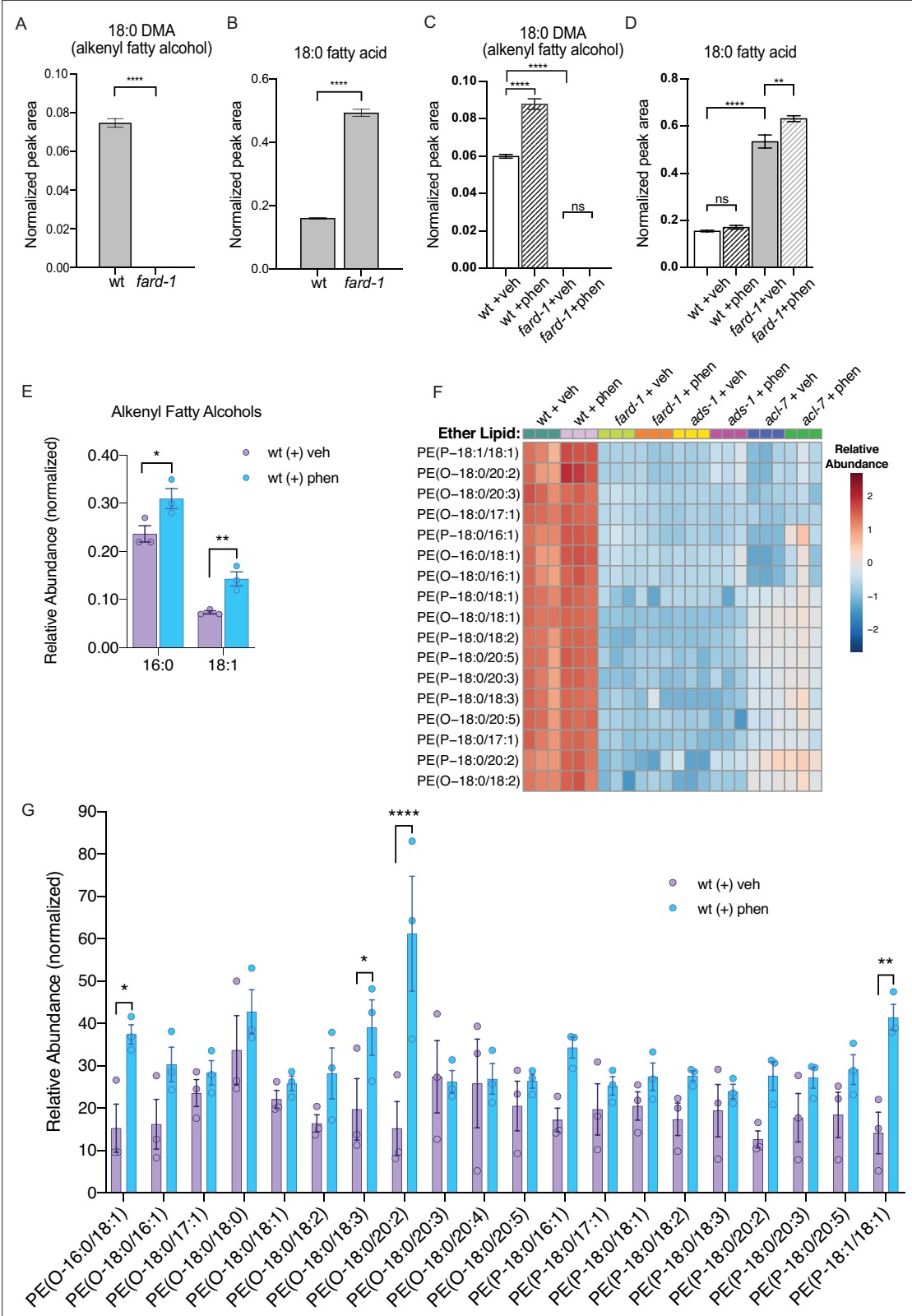

**Figure 2.** Phenformin treatment of *C. elegans* leads to increased abundance of multiple alkyl and alkenyl ether lipids. (**A–B**) Loss-of-function *fard-1* mutants have significant reduction in 18:0 fatty alcohols derivatized from 18-carbon containing alkenyl ether lipids (dimethylacetal [DMA]) by gas chromatography/mass spectrometry (GC/MS) (**A**) and accumulation of the saturated fatty acid stearate (18:0, **B**). (**C**) Wild-type (wt) worms treated with 4.5 mM phenformin display a significant increase in 18:0 DMA relative to vehicle control, indicative of higher levels of alkenyl ether lipids, with levels

*Figure 2 continued on next page*

*Figure 2 continued*

remaining essentially undetectable in *fard-1* mutants on vehicle or drug. (**D**) Phenformin (4.5 mM) treatment does not impact stearate levels in wt worms, however it does result in a greater accumulation of stearate in *fard-1* mutants. For (A–D), **, $p<0.01$; ****, $p<0.0001$, by t-test (**A–B**) or two-way ANOVA (**C–D**), n=3 biological replicates. (**E**) Phenformin (4.5 mM) treatment results in a significant increase in 16:0 DMA and 18:1 DMA in wt worms, relative to vehicle-treated controls *, $p<0.05$; **, $p<0.01$, by multiple t-tests, with two-stage linear step-up procedure of Benjamini, Krieger, and Yekutieli. n=3 biological replicates. (**F**) Heatmap of normalized ether lipid abundance following phenformin treatment in wt *C. elegans* indicates an overall increase in ether lipids relative to vehicle-treated controls, and this shift is absent in ether lipid deficient mutants. All metabolites shown have an FDR adjusted $p<0.05$ by one-way ANOVA followed by Fisher's LSD post hoc testing for wt versus *fard-1, ads-1,* and *acl-7* mutants. (**G**) Liquid chromatography-tandem mass spectrometry (LC-MS) analysis shows that phosphatidylethanolamine-containing ether lipids detected exhibited a general trend toward increased abundance in wild-type worms treated with 4.5 mM phenformin. Four of these ether lipids reached statistical significance: PE(O-16:0/18:1), PE(O-18:0/18:3), PE(O-18:0/20:2), and PE(P-18:1/18:1). Eleven of the ether lipids detected are of the alkyl-type (indicated by 'O' in their name prior to fatty alcohol designation) whereas nine are of the alkenyl-type (plasmalogen, indicated by 'P' in their name prior to the fatty alcohol designation) ether lipids. For (G), *, $p<0.05$; **, $p<0.01$; ****, $p<0.0001$, by multiple t-tests, with multiple hypothesis testing correction by two-stage step-up method of Benjamini, Krieger, and Yekutieli, n=3 biological replicates. See *Figure 2—source data 1* for raw and normalized mass spectrometry data.

The online version of this article includes the following source data and figure supplement(s) for figure 2:

**Source data 1.** Excel file containing raw, normalized, and normalized and $\log_{10}$ transformed mass spectrometry data for phosphatidylethanolamine containing ether lipids detected by liquid chromatography-tandem mass spectrometry (LC-MS/MS).

**Figure supplement 1.** Biguanide treatment modulates abundance of fatty acids in *C. elegans*.

**Figure supplement 2.** FARD-1::RFP localizes to intestinal lipid droplets and peroxisomes and is not positively regulated at the RNA or protein level by phenformin.

to assess the spatial distribution of an integrated FARD-1::RFP reporter *(fard-1 oe3)* in *C. elegans* fed C1-BODIPY-C12 to label lipid droplets (and treated with *glo-4* RNAi to remove BODIPY-positive lysosome-related organelles) (*Hermann et al., 2005*; *Zhang et al., 2010b*; *Zhang et al., 2010a*). We found that FARD-1::RFP fluorescence directly surrounds some, but not all, BODIPY-positive lipid droplets in the worm intestine (*Figure 2—figure supplement 2E*). However, as with peroxisomes, phenformin does not alter the number of lipid droplets that are surrounded by FARD-1 or its distribution around lipid droplets (data not shown). Finally, FARD-1::RFP localizes into web-like structures in the *fard-1(oe3)* reporter that may represent smooth ER versus another cellular tubular vesicular network (*Figure 2—figure supplement 2F*), and this localization is also not altered by biguanide treatment. Thus, the regulation of ether lipid biosynthesis does not appear to be via differential localization of FARD-1.

We next examined expression of mRNAs encoding FARD-1, ACL-7, and ADS-1 following biguanide treatment. Each of these mRNAs decreased or remain unchanged in abundance upon treatment with biguanide via quantitative RT-PCR (*Figure 2—figure supplement 2G–L*), suggesting that ether lipids are not increased in phenformin treatment through a transcriptional mechanism. A parallel decrease in overall levels of FARD-1::RFP protein of *fard-1(oe1)* transgenics was seen with phenformin treatment (*Figure 2—figure supplement 2M*). These seemingly paradoxical data are likely consistent with post-translational negative feedback of ether lipids on the ether lipid biosynthetic pathway, as has been previously reported (*Honsho et al., 2010*).

To affirm that the peroxisome is an essential site of ether lipid production in biguanide action, we disrupted peroxisomal protein targeting and examined phenformin-stimulated lifespan extension. Indeed, either *prx-5* or *prx-19* RNAi impair lifespan extension prompted by phenformin fully or partially, respectively (*Figure 3A–B*). PRX-5 is involved in protein import into the peroxisomal matrix and PRX-19 is involved in proper sorting of proteins for peroxisomal biogenesis. Thus, either disruption of ether lipid biosynthetic machinery or of a principal site of ether lipid biosynthesis impairs phenformin's pro-longevity benefit.

## Fatty acid elongases and desaturases are positive effectors of biguanide-mediated lifespan extension

Most mature ether lipid species contain a fatty acid in the *sn-2* position linked by an ester bond (*Dean and Lodhi, 2018*). The majority of fatty acids conjugated in ether lipids are largely synthesized endogenously in *C. elegans* by fatty acid desaturases and fatty acid elongases (*Perez and Van Gilst, 2008*; *Perez and Watts, 2021*; *Figure 3C*). Thus, we hypothesized that some of these desaturases and elongases may also contribute mechanistically to biguanide-mediated lifespan extension.

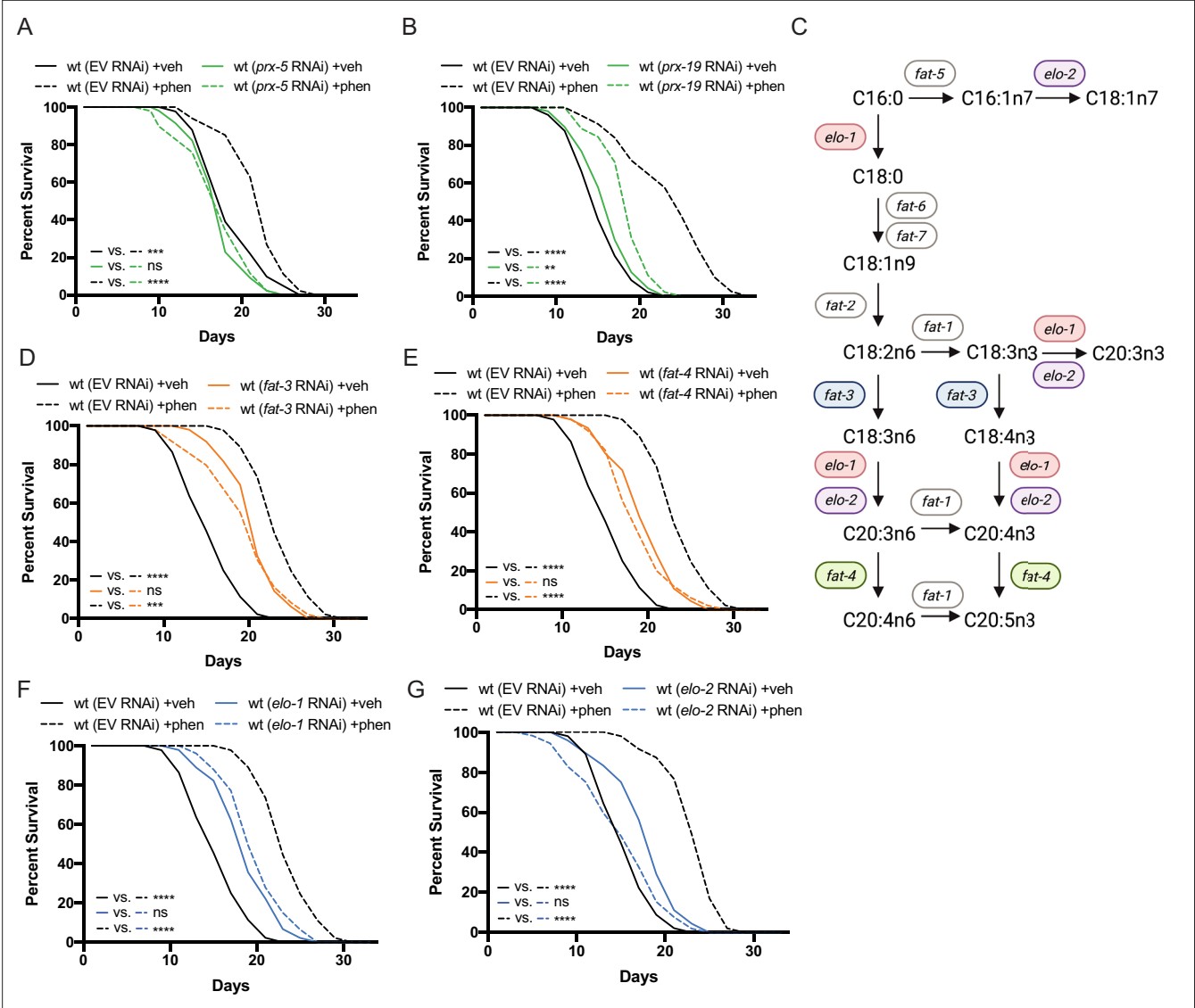

**Figure 3.** Peroxisomal protein import, fatty acid elongases, and fatty acid desaturases are required for the pro-longevity effects of biguanides. (A–B) Knockdown of *prx-5* (A) and *prx-19* (B) by RNA interference (RNAi) eliminates or significantly suppresses phenformin-mediated lifespan extension. (C) Schematic representation of the mono- (MUFA) and polyunsaturated fatty acid (PUFA) synthesis pathway in *C. elegans* (adapted from Figure 1 of *Watts, 2016*). (D–G) RNAi of two fatty acid desaturases (D–E) and two fatty acid elongases (F and G) involved in the synthesis of 18- and 20-carbon PUFAs blunt phenformin-mediated lifespan extension in wild-type worms. Colored symbols for *elo* and *fat* genes (vs. those in black and white) in (C) indicates those that inhibit phenformin lifespan extension when knocked down by RNAi. For (A, B) and (D–G), results are representative of two to three biological replicates. \*\*, p<0.01; \*\*\*, p<0.001; \*\*\*\*, p<0.0001 by log-rank analysis. Note that (D–G) contain the same wild-type controls as they are visualized from the same replicate of the study. See also *Supplementary file 1* for tabular survival data and biological replicates.

Indeed, RNAi knockdown of two fatty acid desaturases and two fatty acid elongases in phenformin-treated *C. elegans* blunted phenformin-stimulated lifespan extension relative to empty vector controls (*Figure 3D–G*). Notably, these four genes all contribute to the production of fatty acids 18–20 carbons in length with three or more double bonds. Although knockdown of fatty acid desaturases and elongases in *C. elegans* results in inherent lifespan extension on vehicle relative to wild-type controls on empty vector RNAi as has been previously reported (*Shmookler Reis et al., 2011*; *Horikawa et al., 2008*), RNAi knockdown of *fat-3*, *fat-4*, *elo-1*, and *elo-2* mitigate phenformin-driven lifespan extension (*Figure 3D–G*). These results suggest the tantalizing possibility that specific fatty acid desaturases and elongases promote biguanide-mediated lifespan extension through contribution of long and

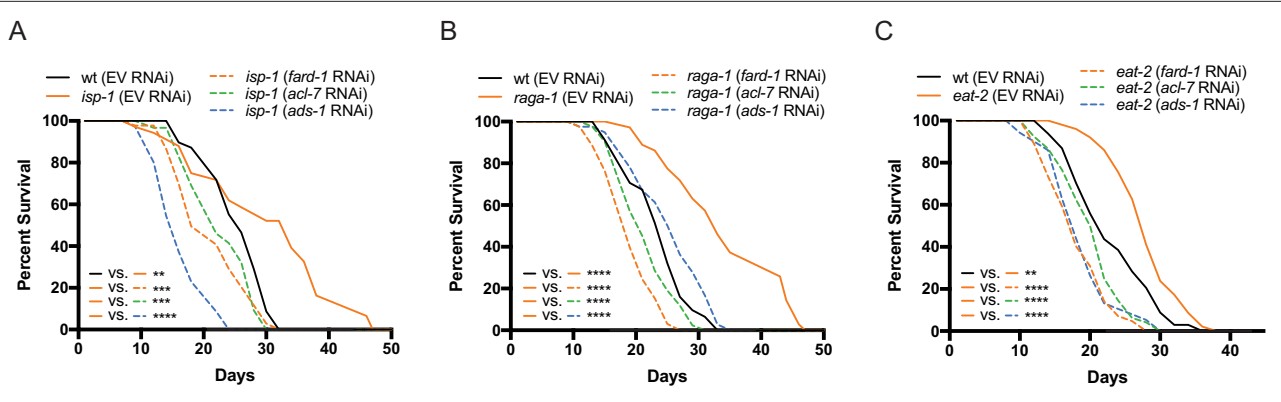

**Figure 4.** Genes involved in ether lipid biosynthesis are required for lifespan extension in multiple longevity paradigms. (**A–C**) *isp-1*, *raga-1*, and *eat-2* mutants display extended lifespan relative to wild-type animals that is dependent upon the three members of the ether lipid biosynthetic pathway. Results are representative of three biological replicates. **, p<0.01; ***, p<0.001; ****, p<0.0001 by log-rank analysis. See also *Figure 4—figure supplement 1* and *Supplementary file 1* for tabular survival data and biological replicates.

The online version of this article includes the following figure supplement(s) for figure 4:

**Figure supplement 1.** Ether lipid biosynthetic genes are not necessary for *daf-2*-dependent lifespan extension.

polyunsaturated fatty acids (PUFAs) to the synthesis of ether lipids, though a mechanistically distinct role is also possible.

## Genes involved in ether lipid biosynthesis are required in multiple longevity paradigms

Given the critical role of ether lipids in the response to biguanides, we hypothesized that these molecules may also play a broader role in diverse longevity paradigms involving metabolic or nutrient-sensing pathways. *C. elegans* mutant strains that exhibit (1) reduced mitochondrial function (*isp-1*), (2) disrupted mTORC1 signaling (*raga-1*), (3) abnormal pharyngeal pumping resulting in a dietary restricted-like state (*eat-2*), or (4) inhibition of insulin/insulin-like growth factor-1 signaling (*daf-2*), all result in extension of lifespan (*Apfeld et al., 2004*; *Curtis et al., 2006*; *Senchuk et al., 2018*; *Schreiber et al., 2010*). To determine whether requirement for the ether lipid biosynthetic machinery in aging generalizes to these other lifespan extension paradigms, we knocked down all three ether lipid biosynthetic enzymes by RNAi in wild-type *C. elegans* and four long-lived genetic mutants: *raga-1*, *isp-1*, *eat-2*, and *daf-2*. Knockdown of *fard-1*, *acl-7*, and *ads-1* by RNAi results in suppression of lifespan extension in *isp-1*, *raga-1*, and *eat-2* mutants (*Figure 4A–C*). However, knockdown of ether lipid synthesis genes by RNAi did not impact lifespan extension in *daf-2* mutants (*Figure 4—figure supplement 1*). Thus, the ether lipid biosynthetic machinery plays a broad role in lifespan extension, and, importantly, does not non-selectively shorten lifespan by making animals generally unfit.

## Overexpression of *fard-1* is sufficient to promote lifespan extension

To determine whether stimulation of ether lipid biosynthesis is sufficient to prompt lifespan extension, we tested the effect of overexpression (*oe*) of the sole *C. elegans* fatty acid reductase that synthesizes fatty alcohols for ether lipid biogenesis, *fard-1,* on lifespan. Strikingly, *fard-1(oe1)* alone significantly extends lifespan (*Figure 5A*). This result is similar in a second, independent *fard-1(oe2)* transgenic line (*Figure 5B*). To confirm that *fard-1(oe)* lifespan extension is dependent upon ether lipid biosynthesis, we knocked down *fard-1*, *acl-7*, and *ads-1* by RNAi in the *fard-1(oe1)* transgenic strain. As predicted, knockdown of three ether lipid biosynthetic enzymes leads to significant suppression of *fard-1(oe1)* lifespan extension (*Figure 5C* and *Figure 5—figure supplement 1A–B*).

We observe that both phenformin treatment and *fard-1* overexpression in *fard-1(oe)* animals non-additively reduce endogenous mRNA expression of *fard-1* by ~40% (*Figure 5D*). This is consistent with our prior observation that mRNA levels of *ads-1*, *acl-7*, and *fard-1* and protein levels of FARD-1 decrease following phenformin treatment (*Figure 2—figure supplement 2G–M*), and invokes end

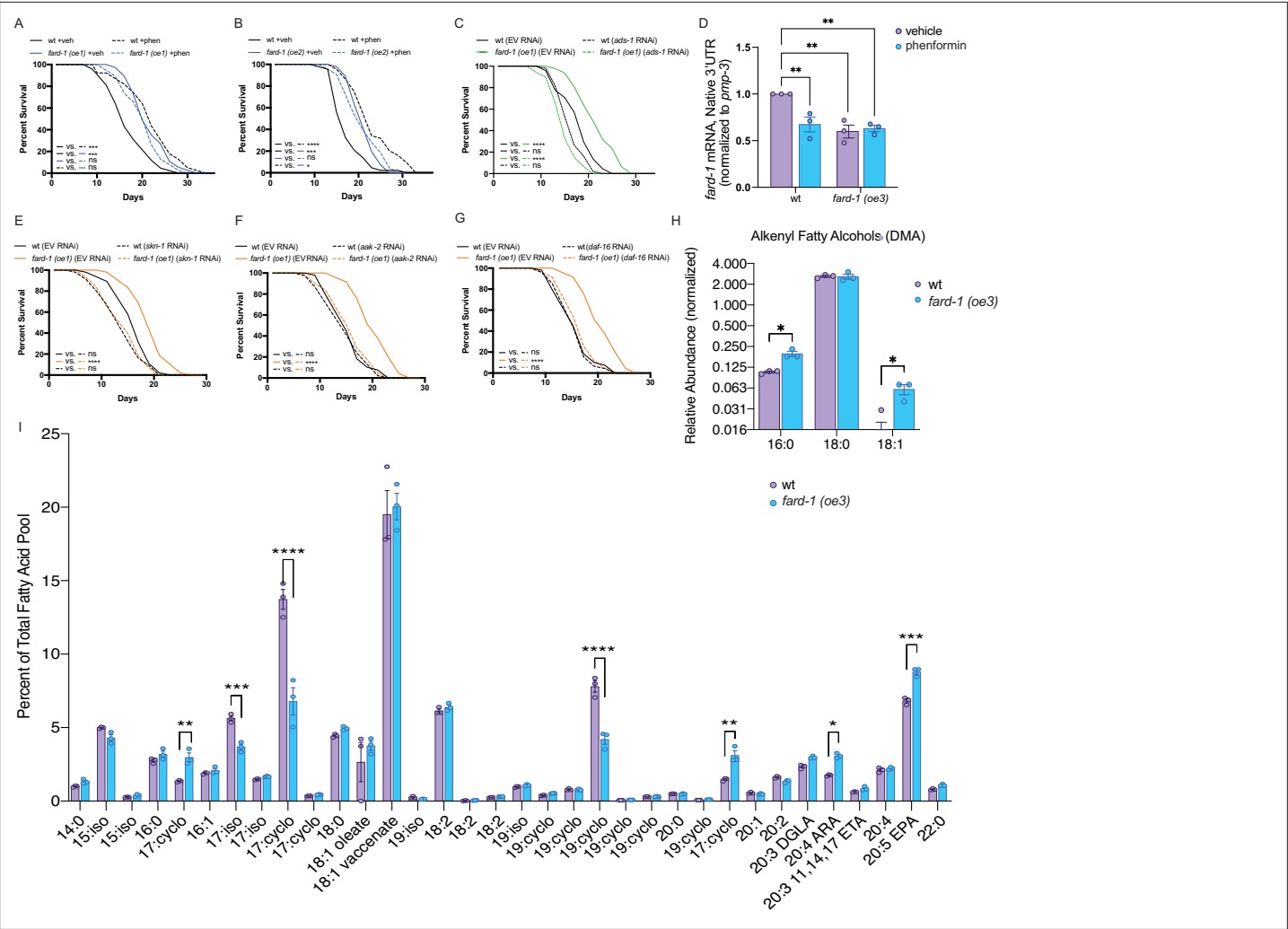

**Figure 5.** *fard-1* overexpression is sufficient to extend lifespan by modulating ether lipid synthesis. (**A–B**) Two independently generated *fard-1* overexpression *(fard-1 oe1* and *fard-1 oe2)* transgenic strains exhibit lifespan extension that is not further extended by concomitant phenformin treatment. (**C**) RNA interference (RNAi) knockdown of *ads-1* fully suppresses *fard-1(oe1)* lifespan extension, indicating that the *fard-1(oe)*-mediated lifespan extension is dependent upon ether lipid synthesis. (**D**) qRT-PCR analysis of wild-type and *fard-1(oe3)* animals treated with vehicle or phenformin until adult day 1 reveals that both biguanide treatment and *fard-1* exogenous overexpression results in an equivalent reduction of native *fard-1* gene expression, as indicated by primers targeting the native 3' UTR of *fard-1*, a sequence not represented in the *fard-1* overexpression transgene (n=3 biological replicates). (**E–G**) RNAi of *skn-1* (**E**), *aak-2* (**F**), and *daf-16* (**G**) suppress *fard-1(oe1)*-mediated lifespan extension. For (A–C) and (E–G), results are representative of two to three biological replicates. *, p<0.05; ***, p<0.001; ****, p<0.0001 by log-rank analysis. Note that (F–G) contain the same wild-type controls as they are visualized from the same replicate of the study. See also ***Figure 4—figure supplement 1*** and ***Supplementary file 1*** for tabular survival data and biological replicates. **, p<0.01 by two-way ANOVA followed by Tukey's multiple comparisons test. (**H**) Worms overexpressing a backcrossed, integrated FARD-1 *(fard-1 oe3)* display a significant increase in 16:0 and 18:1 but not 18:0 alkenyl ether lipids by gas chromatography/ mass spectrometry (GC/MS). (**I**) Comparison of the total fatty acid pool indicates that the polyunsaturated fatty acids 20:4 arachidonic acid (ARA) and 20:5 eicosapentaenoic acid (EPA) are significantly increased in *fard-1* overexpressing *(fard-1 oe3)* worms vs. wild-type animal, while several isomethyl (iso) and cyclopropyl (cyclo) fatty acids change in opposing directions. For (H–I), n=3 biological replicates. *, p<0.05; **, p<0.01; ***, p<0.001; ****, p<0.0001 by multiple t-tests (with multiple hypothesis correction by two-stage step-up method of Benjamini, Krieger, and Yekutieli).

The online version of this article includes the following figure supplement(s) for figure 5:

**Figure supplement 1.** *fard-1* overexpression extends lifespan in a manner dependent upon ether lipid biosynthesis, and not apparently involving ferroptosis.

product negative feedback in the setting of stimulated ether lipid biogenesis, as previously observed in human and animal cellular models (***Honsho et al., 2010***).

To determine whether lifespan extension attributable to *fard-1(oe)* shares genetic dependencies with biguanide-mediated longevity, we independently knocked down *skn-1*/Nrf, *aak-2*/AMPK, and

*daf-16/FoxO* by RNAi in a *fard-1(oe)* background. While *skn-1/Nrf* and *aak-2*/AMPK have previously been demonstrated to be necessary for metformin-stimulated lifespan extension, *daf-16/FoxO* has not (**Onken and Driscoll, 2010**; **Kenyon et al., 1993**). Lifespan extension attributable to *fard-1(oe1)* is suppressed by these three gene knockdowns (**Figure 5E–G**), indicating that it is mechanistically similar, but not identical, to biguanide-mediated lifespan extension (**Cabreiro et al., 2013**; **Onken and Driscoll, 2010**). In aggregate, these results support the notion that ether lipids are an important requirement in multiple, diverse longevity paradigms, and further that *fard-1(oe)* promotes mechanistically distinct lifespan extension in *C. elegans*.

To characterize shifts in ether lipids related to pro-longevity effects, we performed comparative GC-MS-based fatty acid profiling of our integrated *fard-1(oe)* animals. Levels of 16:0 and 18:1 alkenyl ether lipids (indicated by DMAs on GC-MS analysis) are significantly increased in *fard-1(oe3)* transgenic animals versus wild-type worms (**Figure 5H**). By comparison, 18:0 DMA ether lipids were not increased, indicating that the ether lipid pool has both similarities and differences between *fard-1* overexpression and phenformin treatment. Echoing the analysis seen with phenformin treatment, few differences were found in a comparison of the relative abundance of fatty acids within the total lipid pool for *fard-1(oe3)* and wild-type worms (**Figure 5I**). Those exhibiting increases in *fard-1(oe)* include the PUFAs 20:4 arachidonate and 20:5 eicosapentaenoate. This suggests either that PUFAs play a mechanistic role in lifespan extension in *fard-1(oe)* or that they are increased because of longevity-promoting activity of ether lipids.

## Ether lipids do not promote lifespan extension by modulating ferroptosis

Ether lipids have been reported to be protective against ferroptosis, an iron-dependent form of programmed cell death characterized by the accumulation of lipid peroxides (**Zou et al., 2020**; **Perez et al., 2020**). In order to determine whether ether lipids promote longevity downstream of biguanide action by modulating ferroptosis, we knocked down members of the glutathione peroxidase (GPX) family in animals overexpressing integrated *fard-1* (*fard-1 oe3* and *fard-1 oe4*), as has been previously reported to genetically facilitate lipid peroxidation and ferroptosis (**Perez et al., 2020**; **Sakamoto et al., 2014**; **Figure 5—figure supplement 1C–E**). This analysis indicates that *gpx-1* (ortholog of human GPX4) RNAi leads to variable lifespan extension relative to wild-type controls and exhibits non-additive lifespan extension with *fard-1(oe)* (**Figure 5—figure supplement 1C**). Neither *gpx-6* nor *gpx-7* knockdown impacts lifespan extension in *fard-1(oe)* animals (**Figure 5—figure supplement 1D–E**). Further, GPX family RNAi do not negatively impact lifespan extension reproducibly downstream of phenformin (**Figure 5—figure supplement 1F–H**). We conclude that genetic triggers that induce ferroptosis do not impact phenformin-prompted or *fard-1(oe)* lifespan extension, and thus it is unlikely that either extend lifespan by suppressing ferroptosis.

## The ether lipid biosynthetic machinery operates upstream of the stress responsive factor, *skn-1/Nrf*, to enable lifespan extension in response to biguanides

We noted when analyzing FARD-1 protein localization that somatic lipid droplets are generally less numerous in BODIPY-stained phenformin-treated animals vs. vehicle. Indeed, quantitative analysis indicates that intestinal lipid droplets are significantly less numerous following phenformin treatment (in *glo-4* RNAi-treated FARD-1::RFP transgenics *(fard-1 oe3)* fed C1-BODIPY-C12 to label lipid droplets, **Figure 6A**). We previously reported that gain-of-function mutations in the nutrient- and stress-responsive transcription factor *skn-1/Nrf* prompt age-dependent, somatic depletion of fat (Asdf) (**Lynn et al., 2015**; **Nhan et al., 2019**). This, together with early adult decreases in lipid droplet numbers, suggested to us that phenformin may prompt longevity by activating metabolic stress defenses in an *skn-1*-dependent manner. Strikingly, we found that phenformin treatment produces Asdf at day 3 of adulthood, a phenotype that is quantitatively analogous to and non-additive with *skn-1* gain-of-function mutants (**Figure 6B–C**). Compellingly, loss-of-function mutations in any of the three ether lipid biosynthetic genes completely prevent the phenformin-mediated Asdf phenotype (**Figure 6B–C**), suggesting that ether lipids mechanistically connect phenformin to promotion of *skn-1*-dependent pro-longevity metabolic defenses. As expected, *fard-1* overexpressing animals also display an intermediate Asdf phenotype, with moderate enhancement by phenformin treatment (**Figure 6B–C**).

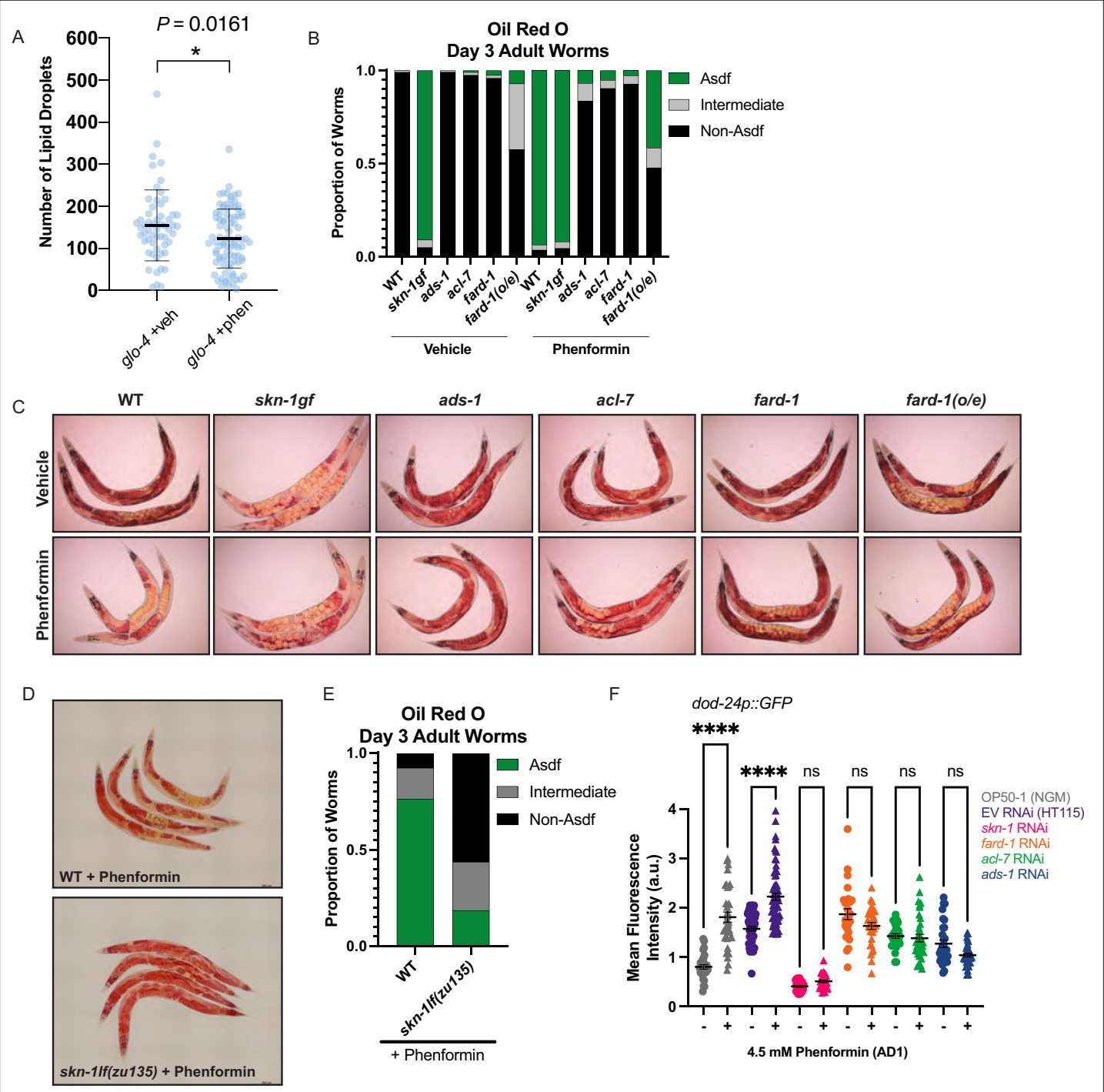

**Figure 6.** Phenformin modulates systemic lipid metabolism through an ether lipid-*skn-1* signaling relay. (**A**) The number of intestinal, C1-BODIPY-C12 labeled lipid droplets are significantly lower in day 1 adult phenformin-treated animals versus vehicle (FARD-1::RFP reporter transgenic [*fard-1 oe3*] worms are also treated with *glo-4* RNA interference (RNAi) to remove BODIPY-positive lysosome-related organelles). n=2 biological replicates. *, p<0.05 by unpaired t-test. (**B–C**) Oil-red-O staining of day 3 adult phenformin-treated wild-type animals indicates that drug treatment leads to age-dependent somatic depletion of fat (Asdf), as previously reported for *skn-1* gain-of-function mutants (*skn-1 gf*), suggesting that phenformin activates Asdf downstream of *skn-1*. Quantification (**B**) indicates that the proportion of Asdf animals is non-additively increased by phenformin treatment in an *skn-1gf* mutant, and that phenformin is no longer able to activate Asdf in three independent ether lipid deficient mutants (*ads-1*, *acl-7*, and *fard-1*). *fard-1* overexpression results in an Asdf phenotype, moderately strengthened by phenformin treatment. For (**B–C**), n=3 biological replicates. (**D–E**) Oil-red-O staining of day 3 adult phenformin-treated wild-type and skn-1lf(zu135) animals reveals that the total loss of *skn-1* function completely abrogates the phenformin-induced Asdf phenotype. Quantification (**E**) reveals that *skn-1lf(zu135)* decreases the proportion of Asdf animals relative to wild-type

*Figure 6 continued on next page*

*Figure 6 continued*

controls treated with phenformin. For (**D–E**), data represent n=3 biological replicates. (**F**) Phenformin treatment induces intestinal expression of *dod-24*, an established SKN-1 response target and innate immune effector, as indicated by increased *dod-24p::GFP* expression, in both OP50-1 and HT115 bacterial diets. RNAi knockdown of *skn-1, fard-1, acl-7, and ads-1* all prevent significant phenformin-mediated induction of *dod-24p::GFP*. Quantification performed with at least 30 animals in each condition (10 animals assayed per replicate for 3 biologically independent experiments). ns, p>0.05; ****, p<0.0001 by two-way ANOVA followed by Tukey's multiple comparisons test.

The online version of this article includes the following figure supplement(s) for figure 6:

**Figure supplement 1.** Biguanides do not activate *gst-4* expression irrespective of bacterial diet.

**Figure supplement 2.** Disruption of bacterial growth and metabolism does not prevent biguanide-mediated induction of ether lipid synthesis.

**Figure supplement 3.** Inactivation of ether lipid machinery disrupts biguanide-mediated lifespan extension independent of effects on bacterial growth or metabolism.

Finally, the Asdf lipid shift evident with phenformin treatment requires *skn-1,* as biguanide-mediated lipid shifts are abrogated in *skn-1* loss-of-function mutant animals (***Figure 6D–E***). In aggregate, these data indicate that ether lipids connect biguanides to activation of metabolic stress defenses and longevity downstream of SKN-1.

Our previous work determined that SKN-1 activates a metabolic stress defense response to drive somatic lipid depletion through enhancing lipid utilization and innate immunity gene expression that opposes canonical oxidative stress responses (***Nhan et al., 2019***). Concordant with the hypothesis that biguanides activate SKN-1 metabolic/innate immune and not oxidative stress defenses, phenformin treatment reduces expression of the canonical oxidative stress response gene *gst-4* irrespective of bacterial diet source (***Figure 6—figure supplement 1A–B***), while reciprocally inducing expression of the innate immune response gene *dod-24* in a manner dependent both upon *skn-1* and ether lipids (***Figure 6F*** and ***Figure 6—figure supplement 1C***). Together with the observation that promotion of lifespan extension by both phenformin and *fard-1(oe)* require *skn-1,* these data suggest that biguanides activate an ether lipid-*skn-1* signaling relay to drive longevity.

## Biguanide-mediated ether lipid synthesis is necessary for a pro-longevity benefit irrespective of bacterial growth or metabolism

Previous studies into the biological action of metformin have suggested that biguanides mediate their lifespan-extending properties in the nematode through alterations in growth and metabolism of their bacterial food source (***Cabreiro et al., 2013***; ***Pryor et al., 2019***). To evaluate whether biguanide effects on ether lipid synthesis are induced through a direct effect on the worm or via alterations in bacterial-host dynamics, we leveraged a robust, established methodology to chemically kill and metabolically inactivate the *C. elegans* OP50-1 food source prior to seeding on nematode growth media (NGM) plates (***Beydoun et al., 2021***). One percent paraformaldehyde (PFA) was identified as the lowest concentration in our hands that completely kills OP50-1 cultures prior to seeding, confirmed through bacterial titer analysis (***Figure 6—figure supplement 2A***). Analysis of wild-type adult day 1 nematodes treated with phenformin on live OP50-1 indicates that biguanide treatment significantly reduces somatic total fatty acid levels (***Figure 6—figure supplement 2B–C***), concordant with our staining data indicating phenformin prompts a low-fat, Asdf state (***Figure 6***). This result was preserved in wild-type animals grown on metabolically inactive PFA-treated OP50-1 as a food source, indicating that biguanides can significantly reduce somatic fatty acid levels irrespective of whether the bacterial food source is live vs. dead and metabolically inactive (***Figure 6—figure supplement 2B–C***). Notably, despite this significant reduction of overall fatty acids, biguanides preferentially protect levels of ether lipid-derived 16:0 DMA and 18:1 DMA, again irrespective of bacterial growth status (***Figure 6—figure supplement 2D–E***). Thus, we conclude that biguanides lead to relative increases in ether lipid levels through direct action in the nematode, rather than through indirect effects in the bacterial food source.

Based on the ability of biguanides to impact the lipid landscape via direct effects on the nematode, we then hypothesized that disruption of ether lipid biosynthesis may also abrogate biguanide-mediated lifespan extension irrespective of bacterial growth and metabolism. First, and in contrast to prior studies suggesting that lifespan requires drug effects on the *Escherichia coli* (***Cabreiro et al., 2013***; ***Pryor et al., 2019***), we noted that under experimental conditions tested, metformin and phenformin both extend lifespan in wild-type animals whether grown on live or PFA-treated *E. coli* OP50-1

(*Figure 6—figure supplement 3A–F*). Consistent with the effect of biguanides on both lifespan and ether lipids being via direct action on the nematode, *ads-1* deficiency completely blunts both metformin and phenformin-mediated lifespan extension irrespective of bacterial food source growth conditions (*Figure 6—figure supplement 3A–F*). In aggregate, these data suggest that biguanides increase proportions of ether lipids and require activated ether lipid machinery to exert pro-longevity benefits through direct drug action on the nematode.

## Discussion

In an unbiased RNAi screen of ~1000 metabolic genes, we identified ether lipid biosynthesis as critical to the longevity-promoting and growth-inhibitory effects of metformin in *C. elegans*. Our results show that the biguanide phenformin promotes lifespan extension by stimulating biogenesis of ether lipids through direct action in the nematode, prompting longevity-promoting metabolic stress defenses mediated by *skn-1*. The broad importance of ether lipids is demonstrated by their requirement in multiple diverse paradigms of lifespan extension. Our findings also indicate that ether lipid modulation through overexpression of *fard-1* is also sufficient to promote longevity. Thus, ether lipids form a heretofore unappreciated lynchpin of lifespan modulation and are sufficient to support healthy aging through multiple central longevity effectors, including *skn-1*.

Differences in ether lipid abundance and composition are correlated with diseases of aging. The uniform lethality associated with human genetic ether lipid deficiency, as in the case of patients diagnosed with RCDP and Zellweger syndrome, has made it difficult to study the role of ether lipids in aging and aging-associated diseases (*Braverman et al., 1997*; *Motley et al., 1997*; *Purdue et al., 1997*; *Itzkovitz et al., 2012*). Nonetheless, observational studies demonstrate decreases in certain plasmalogen species in Alzheimer's disease, suggesting a probable link between ether lipids and aging-related pathologies (*Grimm et al., 2011*; *Goodenowe et al., 2007*; *Han et al., 2001*). Ether lipids have conflicting roles in cancer; while loss of the ether lipid biosynthetic machinery profits cancer cell survival by enhancing resistance to ferroptosis (*Zou et al., 2020*), in other contexts, ether lipid deficiency results in impaired pathogenicity in various human cancer cells (*Benjamin et al., 2013*; *Perez et al., 2020*). Cancer cells generally have higher levels of ether lipids compared to normal cells, leading others to suggest that ether lipids confer pro-survival benefit (*Benjamin et al., 2013*; *Albert and Anderson, 1977*; *Snyder and Wood, 1969*). However, certain ether lipid species have also been reported to have anti-tumor properties (*Jaffrès et al., 2016*; *Arthur and Bittman, 2014*). Thus, in line with the results we present here, it is critical to understand ether lipids in context. Future work will need to focus on the impact of specific ether lipid species rather than the whole class en masse to understand which may play a beneficial versus detrimental role in health.

Studies in long-lived animal models suggest that there is an association between ether lipid content and animal longevity, such as in the naked mole-rat (*Heterocephalus glaber*) (*Mitchell et al., 2007*) and the mud clam *Arctica islandica* (*Munro and Blier, 2012*). Higher plasmalogen levels in naked mole-rat tissues versus mice are speculated to contribute to protection of cellular membranes via a reduction of oxidative stress (*Mitchell et al., 2007*). Similarly, exceptionally long-lived humans harbor higher levels of phosphatidylcholine-derived, short chained alkyl ether lipids and a lower levels of phosphatidylethanolamine-derived longer chained plasmalogens (*Pradas et al., 2019*), but these associations are of unclear functional significance. Although it is clear from work presented here that ether lipid deficiency in *C. elegans* prevents longevity downstream of mitochondrial electron transport chain dysfunction, mTOR deficiency, caloric restriction, and biguanides alike, the precise lipid(s) conferring this activity remains unknown. Each of these longevity paradigms has features of nutrient deficiency, energy stress, or nutrient sensing, so it is possible that ether lipids are at least part of the common effector arm conferring benefit in aging to various forms of metabolic stress. It is particularly interesting that *daf-2* loss-of-function does not require ether lipid biosynthetic machinery and yet has a clear requirement for *skn-1*. These observations suggest the very likely possibilities that (1) it is possible to activate *skn-1* through multiple, parallel mechanisms, only some of which require ether lipids and (2) that there are multiple modes of *skn-1* activation that promote longevity, each of which has distinct transcriptional programs (*Nhan et al., 2019*; *Castillo-Quan et al., 2023*).

Our results suggest that unsaturated fatty acids and phosphatidylethanolamine ether lipids are essential to the health-promoting effects of biguanides. Although we see major shifts in abundance of alkenyl ether lipids, evidence of the necessity of specific ether lipids in biguanide-induced longevity

and for promoting healthy aging awaits the ability to modulate the level of specific ether lipids. Additionally, disruption of ether lipid biosynthesis has been shown to increase the proportion of stearate (18:0) and other saturated fatty acids (*Shi et al., 2016*). Thus, at this time, we cannot rule out the possibility that biguanide-stimulated alterations in ether lipid biosynthesis serves to divert accumulation of lipid species that are detrimental to lifespan, for instance, saturated fatty acids. Nonetheless, considering our finding that ether lipids prompt metabolic stress defenses, this alternative mechanism is less likely. Definitive proof will require a deeper understanding of the regulation of specific steps dictating the synthesis and modification of ether lipids of different fatty alcohol and fatty acid composition.

Based upon our findings, ether lipid synthesis is likely to be regulated post-translationally by biguanide treatment. The demonstrated increases in plasmalogens and specific ether lipids are both consistent with increases in activity of the ether lipid biosynthetic machinery. While we do not understand the mechanism for the increased activity of ether lipid synthesizing enzymes, the decreases in mRNAs for *acl-7*, *ads-1*, and *fard-1* prompted by phenformin treatment and by overexpression of *fard-1* alike invoke negative feedback of the end product(s) of ether lipid biogenesis on transcription of genes encoding ether lipid biosynthetic enzymes. It should be noted that this possibility is consistent with previous work showing that higher levels of ether lipids promote proteasomal degradation of peroxisomal Far1 protein (*Honsho et al., 2010*). Colocalization of the fatty alcohol reductase, FARD-1, with both peroxisomes and lipid droplets is similarly not impacted by biguanides. We cannot rule out the possibility, however, that the exogenous, overexpressed nature of FARD-1::RFP in these experiments may result in a hyperactivated ether lipid biosynthesis state, thereby locking FARD-1::RFP localization in an activated configuration that cannot be further induced with biguanide treatment. Future studies leveraging endogenously tagged FARD-1 animals will be required to resolve this caveat. Finally, further investigation into the precise molecular interactions between FARD-1 protein and other organelles will be required to further understand how FARD-1 and the other ether lipid biosynthetic enzymes are regulated by biguanides and in aging.

Strikingly, our data demonstrate for the first time that ether lipids are required for phenformin to activate metabolic defenses downstream of the stress- and metabolism-responsive transcription factor *skn-1*/Nrf. Phenformin drives age-dependent somatic depletion of fat (Asdf), a phenotype we previously reported upon genetic activation of *skn-1* (*Lynn et al., 2015*; *Nhan et al., 2019*). Based upon our own work, biguanides do not stimulate canonical *skn-1* antioxidant defenses such as *gst-4* expression, in contrast to the subtle effects seen in the existing literature (*Cabreiro et al., 2013*; *Onken and Driscoll, 2010*). Indeed, we observe a significant decrease in *gst-4* expression with phenformin treatment, reciprocally balanced by increased innate immune *dod-24* expression in a manner dependent upon *skn-1* and the ether lipid machinery. We suggest that *skn-1* is uniquely required for metabolic stress defenses downstream of metformin such as Asdf, rather than canonical oxidative or proteostatic defenses. Thus, the requirement for ether lipids in Asdf activation by phenformin confirm that this class of lipids plays a heretofore unappreciated role in a distinct form of *skn-1* activation mimicked by genetic forms of *skn-1* activation that we have previously reported (*Lynn et al., 2015*; *Nhan et al., 2019*).

Lifespan extension resulting from overexpression of *fard-1* shows mechanistic similarities and dissimilarities from biguanide action in longevity. Curiously, and concordant with the idea that ether lipids participate in the activation of *skn-1*, *fard-1* overexpression requires intact *skn-1* and AMPK action. Discordantly, biguanides do not require *daf-16*/FoxO, but *fard-1* overexpression does. There are many possible explanations for this. It could be that *fard-1* overexpression alters the ether lipid landscape in a manner not analogous to biguanide treatment, either in distinct tissues or with regard to different ether lipid molecular species, and that these differences have distinct molecular effectors. Second, the degree of ether lipid alteration may be different following biguanide treatment vs. *fard-1* overexpression. We have not explored whether overexpression of *acl-7* and/or *ads-1* similarly extend lifespan. Further work will be needed to determine the lifespan benefit attributable to augmentation of peroxisomal *acl-7* or *ads-1* activity, and whether any benefit is mechanistically similar to *fard-1* overexpression. Finally, it is highly likely based upon the myriad potential direct and indirect sites of action of biguanides in aging, including but not limited to mitochondria, lysosomes, the nuclear pore complex, mTOR, and AMPK that distinct effector mechanisms are required by biguanides versus *fard-1* overexpression (*Cabreiro et al., 2013*; *Wu et al., 2016*; *Chen et al., 2017*; *Pryor et al., 2019*; *Ma et al., 2022*; *Wheaton et al., 2014*; *Onken and Driscoll, 2010*; *Espada et al., 2020*). In spite of

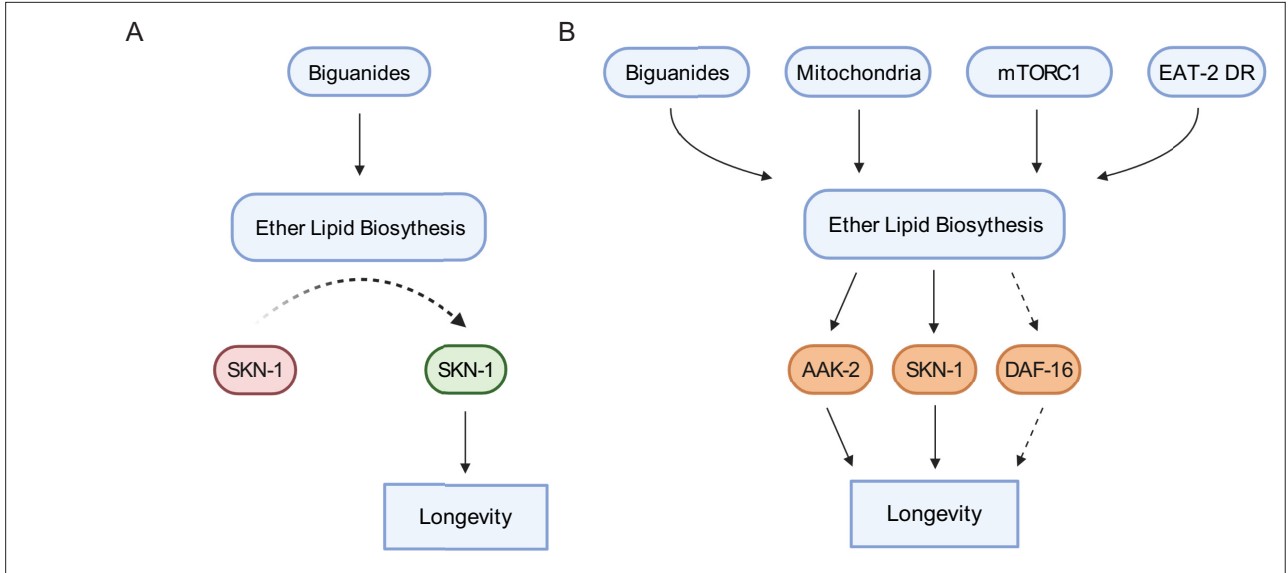

**Figure 7.** Schematic representation for the role of the ether lipid biosynthetic machinery in multiple pro-longevity paradigms. (**A**) Model of ether lipid action in biguanide-prompted lifespan extension. Activation of ether lipid biosynthesis leads to longevity-promoting activity of metabolic stress defenses downstream of the transcription factor *skn-1*. (**B**) Model portraying a broader than previously appreciated role of ether lipids in longevity downstream of biguanides, mitochondrial electron transport inhibition, mTORC1 inhibition, and *eat-2* mutation-mediated dietary restriction (EAT-2 DR). Dashed lines for DAF-16 indicate its requirement for *fard-1* overexpression-, but not biguanide-mediated lifespan extension, suggesting a context-dependent role for *daf-16*/FoxO in mediating pro-longevity outcomes through modulation of ether lipid levels.

the complexity of the biological responses to ether lipids, an opportunity lies ahead: further study of ether lipid roles in aging can provide insights into the full spectrum of signals that favorably impact positive effectors of longevity.

In aggregate, data presented here indicate that ether lipid biosynthesis plays a broader role in aging than previously described. The necessity of the ether lipid machinery in metformin- and phenformin-stimulated lifespan extension and in multiple longevity paradigms indicates that ether lipids serve as a lynchpin through which lifespan is modulated (*Figure 7A–B*). Our demonstration that overexpression of *fard-1* alone results in lifespan extension provides an exciting opportunity to identify ether lipids that promote health and the effector mechanisms through which they act. Finally, these results support the exciting possibility that modulation of ether lipids pharmacologically or even dietarily may provide a new potential therapeutic target in aging and aging-related diseases.

## Materials and methods

### Key resources table

| Reagent type (species) or resource | Designation | Source or reference | Identifiers | Additional information |
|---|---|---|---|---|
| Strain, strain background (*Escherichia coli*) | OP50-1 | Caenorhabditis Genetics Center | RRID: WB-STRAIN:WBStrain00041971 | Standard laboratory stock |
| Strain, strain background (*Escherichia coli*) | HT115(DE3) | Caenorhabditis Genetics Center | RRID: WB-STRAIN:WBStrain00041079 | Background strain for RNAi clones utilized from Ahringer and Vidal Libraries |
| Strain, strain background (*Caenorhabditis elegans*) | Bristol N2 (wt) | Caenorhabditis Genetics Center | RRID: WB-STRAIN:WBStrain00000001 | Standard laboratory wild-type strain |
| Strain, strain background (*Caenorhabditis elegans*) | *fard-1(wa28) [G261D]* | Caenorhabditis Genetics Center | RRID: WB-STRAIN:WBStrain00004025 | BX275 |

*Continued on next page*

*Continued*

| Reagent type (species) or resource | Designation | Source or reference | Identifiers | Additional information |
|---|---|---|---|---|
| Strain, strain background (*Caenorhabditis elegans*) | *acl-7(wa20)* [R234C] | Caenorhabditis Genetics Center | RRID: WS-STRAIN:WBStrain00004024 | BX259 |
| Strain, strain background (*Caenorhabditis elegans*) | *ads-1(wa3)* [G454D] | Caenorhabditis Genetics Center | RRID: WB-STRAIN:WBStrain00004007 | BX10 |
| Strain, strain background (*Caenorhabditis elegans*) | *daf-2(e1370)* | Caenorhabditis Genetics Center | RRID: WB-STRAIN:WBStrain00004309 | CB1370 |
| Strain, strain background (*Caenorhabditis elegans*) | *isp-1(qm150)* | Caenorhabditis Genetics Center | RRID: WB-STRAIN:WBStrain00026672 | MQ989 |
| Strain, strain background (*Caenorhabditis elegans*) | *raga-1(ok701)* | Caenorhabditis Genetics Center | RRID: WB-STRAIN:WBStrain00035849 | VC533 |
| Strain, strain background (*Caenorhabditis elegans*) | *eat-2(da465)* | Caenorhabditis Genetics Center | RRID: WB-STRAIN:WBStrain00005463 | DA465 |
| Strain, strain background (*Caenorhabditis elegans*) | *mgIs43[ges-1p::GFP::PTS1]* | Soukas Laboratory | N/A | MGH48 |
| Strain, strain background (*Caenorhabditis elegans*) | *skn-1(lax188)* | Caenorhabditis Genetics Center | RRID: WB-STRAIN:WBStrain00034420 | *skn-1*gf, SPC168 |
| Strain, strain background (*Caenorhabditis elegans*) | *agIs6[dod-24p::GFP]* | Caenorhabditis Genetics Center | RRID: WB-STRAIN:WBStrain00004921 | CF3556 |
| Strain, strain background (*Caenorhabditis elegans*) | *dvIs19[(pAF15)gst-4p::GFP::NLS]* | Caenorhabditis Genetics Center | RRID: WB-STRAIN:WBStrain00005102 | CL2166 |
| Strain, strain background (*Caenorhabditis elegans*) | *skn-1(zu135)* | Caenorhabditis Genetics Center | RRID: WB-STRAIN:WBStrain00007251 | *skn-1*lf, EU31 |
| Genetic reagent (*Caenorhabditis elegans*) | *alxEx122[fard-1p::FARD-1::mRFP::HA unc-54 3'UTR myo-2p::GFP]* | This study | MGH471 | *fard-1 (oe1)* |
| Genetic reagent (*Caenorhabditis elegans*) | *alxEx135[fard-1p::FARD-1::mRFP::HA unc-54 3'UTR myo-2p::GFP]* | This study | MGH472 | *fard-1 (oe2)* |
| Genetic reagent (*Caenorhabditis elegans*) | *alxIs45[fard-1p::FARD-1::mRFP::HA::unc-54 3'UTR myo-2p::GFP]* | This study | MGH605 | *fard-1 (oe3)*, backcrossed into N2 8× |
| Genetic reagent (*Caenorhabditis elegans*) | *alxIs46[fard-1p::FARD-1::mRFP::HA::unc-54 3'UTR myo-2p::GFP]* | This study | MGH606 | *fard-1 (oe4)*, backcrossed into N2 8× |
| Genetic reagent (*Caenorhabditis elegans*) | *mgIs43[ges-1p::GFP::PTS1]; alxEx122[fard-1p::FARD-1::mRFP::HA::unc-54 3'UTR myo-2p::GFP]* | This study | MGH607 | GFP::PTS1; FARD-1::RFP, prepared by crossing MGH48 into MGH471 |
| Sequence-based reagent | 5'-TGCATGCCTGCAGGTCGACTTTGACAAAAGTTCTGTTGCCG-3' | This study | AS-4524 | Forward primer used to generate *fard-1* overexpression construct |
| Sequence-based reagent | 5'-TTTGGGTCCTTTGGCCAATCGCTTTTTTGAAGATACCGAGAATAATCC-3' | This study | AS-4527 | Reverse primer used to generate *fard-1* overexpression construct |
| Sequence-based reagent | 5'-TGCTGATCGTATGCAGAAGG-3' | This study | *act-1* F | qRT-PCR Primer |

*Continued on next page*

*Continued*

| Reagent type (species) or resource | Designation | Source or reference | Identifiers | Additional information |
|---|---|---|---|---|
| Sequence-based reagent | 5'-TAGATCCTCCGATCCAGACG-3' | This study | *act-1* R | qRT-PCR Primer |
| Sequence-based reagent | 5'-GTTCCCGTGTTCATCACTCAT-3' | This study | *pmp-3* F | qRT-PCR Primer |
| Sequence-based reagent | 5'-ACACCGTCGAGAAGCTGTAGA-3' | This study | *pmp-3* R | qRT-PCR Primer |
| Sequence-based reagent | 5'-ACAAGTCACCAATGGCTCCAC-3' | This study | *fard-1* F | qRT-PCR Primer |
| Sequence-based reagent | 5'-GCTTTGGTCAGAGTGTAGGTG-3' | This study | *fard-1* R | qRT-PCR Primer |
| Sequence-based reagent | 5'-cgatagtgtgtctgttgattgtga-3' | This study | *fard-1* F (Native 3' UTR) | qRT-PCR Primer |
| Sequence-based reagent | 5'-agttattgttgatgagagagtgcg-3' | This study | *fard-1* R (Native 3' UTR) | qRT-PCR Primer |
| Sequence-based reagent | 5'-GTTTATGGCTGGCGTGTTG-3' | This study | *acl-7* F | qRT-PCR Primer |
| Sequence-based reagent | 5'-CGGAGAAGACAGCCCAGTAG-3' | This study | *acl-7* R | qRT-PCR Primer |
| Sequence-based reagent | 5'-GCGATTAACAAGGACGGACA-3' | This study | *ads-1* F | qRT-PCR Primer |
| Sequence-based reagent | 5'-CGATGCCCAAGTAGTTCTCG-3' | This study | *ads-1* R | qRT-PCR Primer |
| Chemical compound, drug | C1-BODIPY-C12 (green) | Invitrogen | Cat#D-3823 | N/A |
| Chemical compound, drug | 5-fluoro-2'-deoxyuridine (FUdR) | Fisher Scientific | Cat#F10705 | N/A |
| Chemical compound, drug | Metformin hydrochloride | MilliporeSigma | Cat#PHR1084 | N/A |
| Chemical compound, drug | Phenformin hydrochloride | MilliporeSigma | Cat#PHR1573 | N/A |
| Commercial assay or kit | Quantitect Reverse Transcription Kit | QIAGEN | Cat#205314 | N/A |
| Commercial assay or kit | Quantitect SYBR Green PCR Reagent | QIAGEN | Cat#204145 | N/A |
| Chemical compound, drug | Levamisole | MilliporeSigma | Cat#L9756 | N/A |
| Software, algorithm | OASIS2 | Structural Bioinformatics Laboratory, POSTECH | https://sbi.postech.ac.kr/oasis2/surv/ | N/A |
| Software, algorithm | MetaMorph | Molecular Devices | https://www.moleculardevices.com/products/cellular-imaging-systems/acquisition-and-analysis-software/metamorph-microscopy | N/A |
| Software, algorithm | Xcalibur (v4.1.31.9) | Thermo Fisher Scientific | Cat#OPTON-30965 | N/A |
| Software, algorithm | QualBrowser (v4.1.31.9) | Thermo Fisher Scientific | Cat#XCALI-97617 | N/A |
| Software, algorithm | MZmine (v2.36) | Open Source | RRID: SCR_012040, http://mzmine.github.io | N/A |
| Software, algorithm | MetaboAnalyst (v5.0) | N/A | https://www.metaboanalyst.ca | N/A |
| Software, algorithm | CellProfiler (v4.2.1) | Broad Institute | https://cellprofiler.org | N/A |
| Software, algorithm | Prism (v9.0) | GraphPad by Dotmatics | https://www.graphpad.com/ | N/A |
| Software, algorithm | Fiji/ImageJ2 (v2.13.1) | NIH | https://imagej.net/software/fiji/ | N/A |

## *C. elegans* genetics

Strains were maintained at 20°C grown on *E. coli* OP50-1 (RRID: WB-STRAIN:WBStrain00041971) for all experiments unless otherwise indicated. The following strains were used in this study: N2 (wt, wild-type strain, RRID: WB-STRAIN:WBStrain00000001), BX275 *fard-1(wa28) [G261D]* (RRID: WB-STRAIN:WBStrain00004025), BX259 *acl-7(wa20)* [R234C] (RRID: WS-STRAIN:WBStrain00004024),

BX10 *ads-1(wa3)* [G454D] (RRID: WB-STRAIN:WBStrain00004007), CB1370 *daf-2(e1370)* (RRID: WB-STRAIN:WBStrain00004309), MQ989 *isp-1(qm150)* (RRID: WB-STRAIN:WBStrain00026672), VC533 *raga-1(ok701)* (RRID: WB-STRAIN:WBStrain00035849), DA465 *eat-2(da465)* (RRID: WB-STRAIN:WBStrain00005463), MGH48 *mgIs43[ges-1p*::GFP::PTS1], SPC168 *skn-1(lax188)* (*skn-1gf*, RRID: WB-STRAIN:WBStrain00034420), CF3556 *agIs6[dod-24p*::GFP] (RRID:WB-STRAIN:WB-Strain00004921), CL2166 dvIs19 [(pAF15)gst-4p::GFP::NLS] (RRID:WB-STRAIN:WBStrain00005102), and EU31 *skn-1(zu135)* (*skn-1lf*, RRID: WB-STRAIN:WBStrain00007251). BX275, BX259, and BX10 strains contain missense mutations that result in loss-of-function of the ether lipid biosynthesis, as previously described (*Shi et al., 2016*). For *fard-1* overexpression, the following strains were generated: MGH471 *alxEx122[fard-1p*::FARD-1::mRFP::HA unc-54 3'UTR *myo-2p*::GFP] (*fard-1 oe1*), MGH472 *alxEx135[fard-1p*::FARD-1::mRFP::HA unc-54 3'UTR *myo-2p*::GFP] (*fard-1 oe2*), MGH605 *alxIs45[fard-1p*::FARD-1::mRFP::HA::unc-54 3'UTR *myo-2p*::GFP] (*fard-1 oe3*), and MGH606 *alxIs46[fard-1p*::FARD-1::mRFP::HA::unc-54 3'UTR *myo-2p*::GFP] (*fard-1 oe4*). All strains for *fard-1* overexpression were backcrossed 8× to wild-type N2 Bristol. For colocalization analysis with peroxisomally targeted GFP, we crossed MGH48 and MGH471 to generate the strain: MGH607 *mgIs43[ges-1p*::GFP::PTS1]; *alxEx122[fard-1p*::FARD-1::mRFP::HA::unc-54 3'UTR *myo-2p*::GFP] (noted in text as GFP::PTS1; FARD-1::RFP).

## Generation of *fard-1 C. elegans* transgenic lines

For FARD-1 expression, the entire genomic sequence of the *fard-1* locus (3659 bp), including introns and exons, plus 4910 bp of promoter were amplified and cloned into a modified Fire vector driving *fard-1* fused to mRFP and an HA epitope tag at the C-terminus. The following cloning primers were used:

F: 5'-TGCATGCCTGCAGGTCGACTTTGACAAAAGTTCTGTTGCCG-3' and
R: 5'-TTTGGGTCCTTTGGCCAATCGCTTTTTTGAAGATACCGAGAATAATCC-3'.

The *fard-1* overexpression construct was injected at 10 ng/µL (*alxEx122*) and 18 ng/µL (*alxEx135*) into the gonad of wild-type adult animals with salmon sperm DNA as a carrier and 1.5 ng/µL *myo-2p*::GFP as a co-injection marker. *alxEx122* was subsequently integrated by UV irradiation and 8× backcrossed to wild-type N2 Bristol to obtain MGH605 and MGH606.

## RNAi assays

RNAi clones were isolated from a genome-wide *E. coli* RNAi library (generated in strain HT115(DE3), RRID: WB-STRAIN:WBStrain00041079), sequence verified, and fed to animals as described (*Kamath and Ahringer, 2003*). RNAi feeding plates (6 cm) were prepared using a standard NGM recipe with 5 mM isopropyl-β-D-thiogalactopyranoside and 200 µg/mL carbenicillin. RNAi clones were grown for 15 hr in Luria Broth (LB) containing 100 µg/mL carbenicillin with shaking at 37°C. The stationary phase culture was then collected, concentrated through centrifugation, the supernatant was discarded, and the pellet was resuspended in LB to 20% of the original culture volume; 250 µL of each RNAi clone concentrate was added to RNAi plates and allowed to dry at least 24 hr prior to adding biguanide. Drug treatment was added to seeded RNAi plates and allowed to dry at least 3 hr before adding worms.

## Longevity assays

Lifespan analysis was conducted at 20°C, as previously described (*Soukas et al., 2009*). Briefly, synchronized L1 animals were seeded onto NGM (for mutant treatment) or RNAi plates (for RNAi) and allowed to grow until the L4 to YA transition stage. On day 0 of adulthood as indexed in the figure legend, ~50–60 L4/YA worms per plate (unless otherwise noted) were transferred onto fresh NGM or RNAi plates. These NGM and RNAi plates were supplemented with 30 µM and 100 µM FUdR to suppress progeny production, respectively. For biguanide treatment, about ~55–60 synchronized L1 animals (unless otherwise noted) were seeded onto plates containing 50 mM metformin or 4.5 mM phenformin. Based upon power calculations for log-rank analysis, minimum N of 50 (per group) was chosen to satisfy α=0.05, β=0.2, and effect size = 20% difference in lifespan (*Petrascheck and Miller, 2017*). At the L4/YA stage, these worms were transferred to plates containing biguanide treatment and FUdR for the remainder of their life. For experiments performed without the use of FUdR, animals

were transferred to freshly seeded RNAi and drug supplemented plates every 2 days between day 0 and day 10 of adulthood, ensuring no crossover contamination of progeny or laid eggs on the lifespan plates until the animals cease the reproductive stage. Dead worms were counted every other day, and scoring investigators were blinded as to the experimental group/treatment until the conclusion of each experiment. All lifespans performed include same-day N2 wild-type (wt) controls examined simultaneously with experimental test animals in each study. Statistical analysis was performed with online OASIS2 resources (*Han et al., 2016*).

## Body size determination of *C. elegans*

We measured worm body size in response to biguanide treatment by imaging as previously described (*Wu et al., 2016*). Egg prep synchronized wild-type worms were treated with empty vector (L4440) or ether lipid biosynthesis machinery RNAi and treated with vehicle (ddH$_2$O) or 160 mM metformin. After ~65–70 hr, worms were transferred into a 96-well plate, washed 3× with M9, and paralyzed in M9 buffer with 1 mg/mL levamisole (L9756-10G, Sigma-Aldrich). Once immobilized, brightfield imaging was performed at ×5 magnification on a Leica DM6000 microscope within 5 min of transferring to a 96-well Teflon imaging slide. We determined the maximal, longitudinal cross-sectional area of the imaged *worms* by using MetaMorph software for a minimum of ~80 animals per condition in each experiment. Results of a single experiment are shown. Each experiment was performed at least twice, and results were consistent between experiments.

## GC/MS lipidomics

Lipid extraction and GC/MS of extracted, acid-methanol-derivatized lipids was performed as described previously (*Pino and Soukas, 2020*; *Pino et al., 2013*). Briefly, 5000 synchronous mid-L4 animals were sonicated with a probe sonicator on high intensity in a microfuge tube in 100–250 µL total volume. Following sonication, lipids were extracted in 3:1 methanol:methylene chloride following the addition of acetyl chloride in sealed borosilicate glass tubes, which were then incubated in a 75°C water bath for 1 hr. Derivatized fatty acids and fatty alcohols were neutralized with 7% potassium carbonate, extracted with hexane, and washed with acetonitrile prior to evaporation under nitrogen. Lipids were resuspended in 200 µL of hexane and analyzed on an Agilent GC/MS equipped with a Supelcowax-10 column as previously described (*Pino and Soukas, 2020*). Fatty acids and alcohols are indicated as the normalized peak area of the total of derivatized fatty acids and alcohols detected in the sample. Based upon power calculation for pairwise comparison, a minimum n of 3 biological replicates (per group) was chosen to satisfy α=0.05, β=0.2, and effect size = 50% with σ=20%. Analyses were blinded to the investigator conducting the experiment and mass spectrometry calculations until the conclusion of each experiment when aggregate statistics were computed.

## LC/MS-MS lipidomics

Wild-type, *fard-1*, *acl-7*, and *ads-1* worm mutants were collected using conditions that enabled our reported longevity phenotypes. Briefly, collection for LC/MS-MS processing comprised of three replicates of these four strains that were independently treated with vehicle (ddH$_2$O) and 4.5 mM phenformin on 10 cm NGM plates. Based upon power calculations, as for GC/MS, a minimum n of 3 biological replicates (per group) was chosen to satisfy α=0.05, β=0.2, and effect size = 50% with σ=20%, though the power is only expected to hold for the first significant difference detected. Analyses were blinded to the investigator conducting the experiment and mass spectrometry calculations until the conclusion of each experiment when aggregate statistics were computed. A total of ~6000 animals (2×10 cm plates, 3000 worms per plate) were utilized per sample. These worms were washed with M9 (4×), concentrated into 200 µL of M9, and then flash frozen with liquid nitrogen in 1.5 mL Eppendorf microcentrifuge tubes. Worm pellets were transferred to 2 mL impact resistant homogenization tubes containing 300 mg of 1 mm zirconium beads and 1 mL of 90:10 ethanol:water. Using a Precellys 24 tissue homogenizer, samples were homogenized in three 10 s cycles at 6400 Hz followed by 2 min of sonication. Samples were then placed at –20°C for 1 hr to facilitate protein precipitation. Samples were transferred to 1.5 mL microfuge tubes and centrifuged at 14,000 × *g* for 10 min at 4°C. After centrifugation, 120 µL of supernatant was dried in vacuo and resuspended in 120 µL of 80:20 methanol:water containing internal standards 1 ng/µL CUDA and 1 ng/µL MAPCHO-12-d38. Lipidomic data was acquired by injecting 20 µL of sample onto a Phenomenex Kinetex F5 2.6 µm

(2.1×100 mm) column at 40°C and flowing at 0.35 mL/min. Metabolites were eluted using (A) water containing 0.1% formic acid and (B) acetonitrile:isopropanol (50:50) containing 0.1% formic acid using the following gradient: 0% B from 0 to 1 min, 0–50% B from 1 to 6 min, 50–100% B from 6 to 17 min, and 100% B hold from 17 to 20 min. Compounds were detected using a Thermo Scientific QExactive Orbitrap mass spectrometer equipped with a heated electrospray ionization source operating in positive and negative ion mode with the following source parameters: sheath gas flow of 40 units, aux gas flow of 15 units, sweep gas flow of 2 units, spray voltage of ±3.5 kV, capillary temperature of 265°C, aux gas temp of 350°C, S-lens RF at 45. Data was collected using an MS1 scan event followed by four DDA scan events using an isolation window of 1.0 m/z and a normalized collision energy of 30 arbitrary units. For MS1 scan events, scan range of m/z 100–1500, mass resolution of 17.5K, AGC of $1e^6$ and inject time of 50 ms was used. For tandem MS acquisition, mass resolution of 17.5K, AGC $5e^5$ and inject time of 80 ms was used. Data was collected using Thermo Xcalibur software (version 4.1.31.9) and analyzed using Thermo QualBrowser (version 4.1.31.9) as well as MZmine 2.36.

## Statistical analysis of metabolomics data

All visualization and significance testing of metabolomics was conducted using the MetaboAnalyst 5.0 package (*Pang et al., 2021*). Mass integration values for 9192 compounds were extracted from full-scan LC-MS/MS measurements of L4 to young adult (YA) transition wild-type (N2 Bristol), *ads-1(wa3)*, *acl-7(wa20)*, and *fard-1(wa28)* animals treated from L1 hatch with vehicle, 4.5 mM phenformin, or 50 mM metformin. Missing and zero values in the data matrix were imputed via replacement with 1/5th of the minimum positive value for each variable. Abundance values were subsequently filtered based on interquartile range (reducing the compound list to the 2500 most variable compounds), and $log_{10}$ transformed. Quantile normalization was then performed, followed with division by the standard deviation of each variable (auto-scaling). Normalized abundance values for each metabolite were then extracted based upon MS/MS signatures for phosphatidylethanolamine ether lipids and assessed for statistical significance via one-way ANOVA followed by false discovery rate (FDR) control using the Benjamini-Hochberg method (*Benjamini and Hochberg, 1995*). Post hoc testing was then performed using Fisher's LSD to evaluate pairwise comparison significance. Metabolites were considered differentially abundant in any one condition with an FDR controlled p-value <0.05. The top 25 metabolites across treatment (ranked by ANOVA f statistic and FDR value) were visualized using a heatmap of Euclidean distance measurements, with Ward clustering of samples and normalized compound abundances included. All mass integration values for identified phosphatidylethanolamine containing ether lipids, normalized abundance values, and log-transformed, normalized abundance values are included in this manuscript as *Figure 2—source data 1*. These same data have been made publicly available and can be found at Dryad.

## Quantitative RT-PCR

To assess changes in mRNA levels of *fard-1*, *acl-7*, and *ads-1* in response to biguanide treatment, we used quantitative RT-PCR as previously described (*Wu et al., 2016*). Briefly, synchronized wild-type N2 (wt) or *fard-1 (oe3)* L1 animals were seeded onto OP50-1 NGM plates containing vehicle (ddH₂O), 50 mM metformin, or 4.5 mM phenformin. ~1600 worms were collected from four 6 cm plates per replicate, per condition (with no more than 400 worms seeded per plate to prevent overcrowding). n=3 biological replicates. Worms were collected at the L4 to YA transition (for wild-type analysis) or at adult day 1 (for *fard-1 (oe3)* analysis) using M9 buffer and washed an additional 3×, allowing worms to settle by gravity between washes. Total RNA was extracted using TRIzol and phenol-chloroform extraction. Reverse transcription was performed with the Quantitect Reverse Transcription kit (QIAGEN). qRT-PCR was conducted in triplicate using Quantitect SYBR Green PCR reagent (QIAGEN) following the manufacturer's instructions on a Bio-Rad CFX96 Real-Time PCR system (Bio-Rad). If not processed immediately, worms were flash frozen in liquid nitrogen and kept in −80°C until RNA preparation. The sequences for primer sets used in *C. elegans* are:

> act-1:
> F: 5'-TGCTGATCGTATGCAGAAGG-3' and
> R: 5'-TAGATCCTCCGATCCAGACG-3'
> pmp-3:
> F: 5'-GTTCCCGTGTTCATCACTCAT-3' and

R: 5'-ACACCGTCGAGAAGCTGTAGA-3'
fard-1 (spanning Exons 5–6, *Figure 2—figure supplement 2G and J*) :
F: 5'-ACAAGTCACCAATGGCTCCAC-3' and
R: 5'-GCTTTGGTCAGAGTGTAGGTG-3'
*fard-1* (native 3' UTR, *Figure 5D*):
F: 5'-cgatagtgtgtctgttgattgtga-3' and
R: 5'-agttattgttgatgagagagtgcg-3'
acl-7:
F: 5'-GTTTATGGCTGGCGTGTTG-3' and
R: 5'-CGGAGAAGACAGCCCAGTAG-3'
ads-1:
F: 5'-GCGATTAACAAGGACGGACA-3' and
R: 5'-CGATGCCCAAGTAGTTCTCG-3'.

Expression levels of tested genes were presented as normalized fold changes to the mRNA abundance of *act-1* or *pmp-3* for *C. elegans* by the ΔΔCt method.

## *fard-1* overexpression reporter fluorescence intensity analysis

To assess changes in levels of fluorescent FARD-1 protein in response to biguanide treatment, we used the strain MGH471 *alxEx122*[*fard-1p*::FARD-1::mRFP::HA *unc-54* 3'UTR *myo-2p*::GFP] *(fard-1 oe1)*. In brief, egg prep synchronized L1 FARD-1::RFP transgenic worms were treated with vehicle (ddH$_2$O) or 4.5 mM phenformin, paralyzed with 1 mg/mL of levamisole, and then imaged in 96-well format with a Leica DM6000 microscope outfitted with a mCherry filter set and MMAF software. These imaging experiments were carried out in biological triplicate with ~10 animals imaged per replicate. Images were qualitatively assessed to obtain conclusions and results were consistent between independent replicates.

## Colocalization analysis of FARD-1::RFP and peroxisomally targeted GFP

Colocalization of GFP and RFP expression in vehicle- or phenformin-treated MGH607 was performed by Coloc2 (Fiji) on images taken on a Leica Thunder microscopy system. Since FARD-1::RFP in MGH607 is exogenously expressed, we performed 3 hr egg lays with ~30 gravid hermaphrodites expressing both GFP::PTS1 and FARD-1::RFP to synchronize L1s. The eggs were treated with vehicle (ddH$_2$O) or 4.5 mM phenformin immediately after gravid hermaphrodites were removed, dried in a laminar flow hood, and allowed to incubate at 20°C until the worms were YA/early day 1 adults. To prepare for imaging, only worms expressing both GFP::PTS1 and FARD-1::RFP were picked onto slides containing dried 2% agar pads, immobilized in ~5 µL of 2.5 mM levamisole solution and covered with a coverslip. Images of the upper, mid, and lower intestine were taken for 30 individual worms per condition (15 worms per replicate for two biological replicates). We generated Pearson's r values to assess the extent to which intestinal RFP and GFP overlap in each region of all samples. All Pearson's r values were combined to generate four individual averages (one per condition) to perform an unpaired t-test.

## Lipid droplet analysis

The strain MGH605 *alxIs45*[*fard-1p*::FARD-1::mRFP::HA::*unc-54* 3'UTR *myo-2p*::GFP] *(fard-1 oe3)* was used for this analysis. Preparation of worms for imaging was similar to our longevity assays but modified to incorporate staining of lipid droplets. Briefly, 6 cm RNAi plates were seeded with 250 µL bacteria expressing *glo-4* RNAi [5×] and allowed to incubate for 24 hr at 20°C. One µM of green C1-BODIPY-C12 (D-3823, Invitrogen) diluted in 100 µL 1× phosphate buffer saline (PBS, pH 7.2) was then added to the RNAi bacteria lawn as in *Soukas et al., 2009*. The plates were immediately dried in a dark laminar flow hood, wrapped in aluminum foil to prevent photobleaching, and allowed to incubate at 20°C for 24–48 hr. These plates were treated with vehicle (ddH$_2$O) or 4.5 mM phenformin as mentioned previously (while kept away from light). Egg prep synchronized worms were dropped onto plates and grown to day 1 adult stage. To prepare for confocal imaging, animals were rapidly picked onto slides containing dried 2% agar pads, immobilized in ~5 µL of 2.5 mM levamisole solution and covered with a coverslip. Lipid droplets were imaged by Zeiss LSM 800 Airyscan within 5 min of slide placement. Z-stacked images were obtained for the intestine near the tail end of 14 *glo-4*, vehicle-treated and 19 *glo-4*, phenformin-treated worms (two biological replicates per condition). Five planes

were extracted (planes 1, 2, 4, 5, and 9) using ImageJ for all samples. For lipid droplet counting, quantification was performed using CellProfiler 4.2.1 (*Stirling et al., 2021*) where lipid droplets were identified as primary objects. The min/max range for typical object diameters was 3–67 pixels, and those objects outside of the diameter range were discarded. Planes were excluded entirely if the pipeline did not accurately capture individual lipid droplets for the vast majority of objects.

## Oil-red-O staining

Oil-red-O (ORO) fat staining was conducted as outlined in *Stuhr et al., 2022*, In brief, worms were synchronized by bleach prep and allowed to hatch overnight for a synchronous L1 population. The next day, worms were dropped onto plates seeded with bacteria with or without phenformin and raised to 120 hr (day 3 adult stage). Worms were washed off plates with PBST, then rocked for 3 min in 40% isopropyl alcohol before being pelleted and treated with ORO in diH$_2$O for 2 hr. Worms were pelleted after 2 hr and washed in PBST for 30 min before being imaged at ×5 magnification with the DIC filter on the Zeiss Axio Imager Erc color camera. A minimum of 200 worms in total (across three independent biological replicates) were assessed per condition for final quantification and evaluation.

## Generation of metabolically inactive *E. coli* for lipidomic and lifespan studies

PFA killing of OP50-1 *E. coli* was performed as previously described with slight modifications (*Beydoun et al., 2021*). 50 mL aliquots of OP50-1 liquid cultures grown overnight in LB media supplemented with 25 µg/mL streptomycin were dispensed into 250 mL Erlenmeyer flasks. Either 1× PBS (Life Technologies) for mock treatment or 4% PFA (Sigma-Aldrich) diluted in 1× PBS was added to each flask for a final concentration of 1% (vol/vol). Bacteria were then shaken in 37°C at 210 rpm for 2 hr to enable PFA inactivation. Cultures were then aseptically transferred into 50 mL conical centrifuge tubes, and then washed 6× with sterile PBS to remove residual PBS or PFA solution. After the final wash, bacterial pellets were then 10× concentrated in LB media supplemented with 25 µg/mL streptomycin, and 300 µL seeded onto freshly prepared NGM plates. Plates were allowed to dry for 2 days prior to use for GC/MS or lifespan analyses. A standard culture of OP50-1 grown overnight was similarly 10× concentrated and seeded as a 'live OP50-1' control to compare to mock-treated and PFA-treated bacterial conditions. Bacterial titer calculations were performed as previously described (*Beydoun et al., 2021*), removing an 10 µL aliquot of culture prior to plate seeding, diluting 10 times in 10-fold serial dilutions, and subsequently dispensing the 100 µL dilutions onto LB agar plates and aseptically spread across the surface evenly. Plates were incubated at 37°C overnight before counting colonies for colony forming units and titer calculations.

## Asdf quantification

ORO-stained worms were placed on glass slides and a coverslip was placed over the sample. Worms were scored, as previously described (*Stuhr et al., 2022*). Worms were scored and images were taken with the Zeiss Axio Imager Erc color camera at ×5 magnification. Fat levels of worms were placed into three categories: non-Asdf, intermediate, and Asdf. Non-Asdf worms display no loss of fat and are stained dark red throughout most of the body (somatic and germ cells). Intermediate worms display significant fat loss from the somatic tissues, with portions of the intestine being clear, but ORO-stained fat deposits are still visible (somatic<germ cells). Asdf worms had most, if not all, observable somatic fat deposits depleted (germ cells only).

## Fluorescence reporter imaging and quantification

GFP imaging of CF3556 *agIs6*[*dod-24p*::GFP] and CL2166 *dvIs19* [(*pAF15)gst-4p::GFP::NLS*] animals was performed using a fully automated, high-speed fluorescence Leica THUNDER 3D imaging station at ×5 magnification. Egg prep synchronized *dod-24p::GFP* or *gst-4p::NLS::GFP* animals were dropped onto NGM plates seeded with OP50-1, or RNAi plates seeded with L4440 (EV), *skn-1*, *fard-1*, *acl-7*, or *ads-1* HT115 RNAi clones, treated with vehicle (water) or 4.5 mM phenformin, and grown to adult day 1 stage. Animals were rapidly picked onto slides containing dried 2% agar pads, immobilized in ~5 µL of 2.5 mM levamisole solution, and covered with a coverslip. Quantification was performed using ImageJ/Fiji (*Schindelin et al., 2012*), in which at least 10 animals per condition per replicate were randomly polygon traced, collected into an ROI manager, and measured for mean fluorescence

intensity (MFI). MFI values per condition per replicate (n=3) were aggregated using Prism 9 (GraphPad) for visualization and subsequent statistical analysis.

## Quantification and statistical analysis

Unless otherwise indicated, the statistical differences between control and experimental groups were determined by two-tailed Student's t-test (two groups), one-way ANOVA (more than two groups), or two-way ANOVA (two independent experimental variables), with corrected p-values <0.05 considered significant. Analyses conducting more than two comparisons were always corrected for multiple hypothesis testing. The log-rank test was used to determine significance in lifespan analyses using online OASIS2 (https://sbi.postech.ac.kr/oasis2/).

## Acknowledgements

We thank Talia Hart, Dr. Gary Ruvkun, Dr. Eric Greer, and Dr. Keith Blackwell for discussions and constructive criticisms. This work was funded by NIH/NIA Grants R01AG058259 and R01AG69677 (to AAS) and R01AG058610 (to SPC), by the Weissman Family MGH Research Scholar Award (to AAS), by an NSF GRFP Award 1000253984 (to LC), and by NIH/NIAID R01AI130289 (to RPW), and by IRACDA NIH Grant K12GM106996 (to LC). Thanks to the University of Southern California and Buck Institute Nathan Shock Center (P30AG068345) for providing core services and support. Thanks to the NIH/NIDDK-funded NORC of Harvard (P30DK040561) and the NIH/NIDDK-funded Boston-Area DERC (P30DK057521) for core services. Some strains were provided by the CGC, funded by the NIH Office of Research Infrastructure Programs (P40OD010440), and the *C. elegans* Knockout Consortium. *Figures 1A, 3C and 7* were created with BioRender.com.

## Additional information

### Funding

| Funder | Grant reference number | Author |
|---|---|---|
| National Institutes of Health | R01AG058259 | Alexander A Soukas |
| National Institutes of Health | R01AG69677 | Alexander A Soukas |
| National Institutes of Health | R01AG058610 | Sean P Curran |
| National Science Foundation | Graduate Research Fellowship Program Award 1000253984 | Lucydalila Cedillo |
| National Institutes of Health | R01AI130289 | Read Pukkila-Worley |
| National Institutes of Health | K12GM106996 | Lucydalila Cedillo |
| University of Southern California and Buck Institute Nathan Shock Center | P30AG068345 | Sean P Curran |
| Nutrition Obesity Research Center at Harvard | P30DK040561 | Alexander A Soukas |
| NIH/NIDDK-funded Boston-Area DERC | P30DK057521 | Alexander A Soukas |

The funders had no role in study design, data collection and interpretation, or the decision to submit the work for publication.

## Author contributions
Lucydalila Cedillo, Conceptualization, Formal analysis, Validation, Investigation, Visualization, Methodology, Writing – original draft, Writing – review and editing; Fasih M Ahsan, Conceptualization, Formal analysis, Validation, Investigation, Visualization, Methodology, Writing – review and editing; Sainan Li, Jeramie Watrous, Validation, Investigation, Methodology, Writing – review and editing; Nicole L Stuhr, Sean P Curran, Investigation, Visualization, Methodology, Writing – review and editing; Yifei Zhou, Yuyao Zhang, Adebanjo Adedoja, Investigation, Methodology, Writing – review and editing; Luke M Murphy, Validation, Investigation, Writing – review and editing; Armen Yerevanian, Sinclair Emans, Khoi Dao, Investigation, Writing – review and editing; Zhaozhi Li, Sudeshna Das, Formal analysis, Visualization, Writing – review and editing; Nicholas D Peterson, Read Pukkila-Worley, Formal analysis, Investigation, Writing – review and editing; Mohit Jain, Formal analysis, Investigation, Methodology, Writing – review and editing; Alexander A Soukas, Conceptualization, Formal analysis, Supervision, Funding acquisition, Validation, Investigation, Visualization, Methodology, Writing – original draft, Writing – review and editing

## Author ORCIDs
Fasih M Ahsan http://orcid.org/0000-0001-8031-7056
Sainan Li http://orcid.org/0000-0002-1880-6294
Nicole L Stuhr https://orcid.org/0000-0003-2537-7114
Yifei Zhou http://orcid.org/0000-0003-0088-6262
Luke M Murphy http://orcid.org/0000-0002-2784-6255
Nicholas D Peterson https://orcid.org/0000-0003-4157-8119
Read Pukkila-Worley http://orcid.org/0000-0001-5340-8294
Sean P Curran https://orcid.org/0000-0001-7791-6453
Alexander A Soukas http://orcid.org/0000-0002-9100-2436

## Decision letter and Author response
Decision letter https://doi.org/10.7554/eLife.82210.sa1
Author response https://doi.org/10.7554/eLife.82210.sa2

# Additional files

## Supplementary files
• Supplementary file 1. Tabular and survival data including three biological replicates (unless otherwise noted) are shown for lifespan experiments related to *Figures 1 and 3–5*, *Figure 1—figure supplement 1*, *Figure 1—figure supplement 2*, *Figure 4—figure supplement 1*, *Figure 5—figure supplement 1*, and *Figure 6—figure supplement 3*. Data present a summary of the conditions tested which, if applicable, include: (1) drug treatment with vehicle control and 4.5 mM phenformin or 50 mM metformin and/or (2) RNAi treatment to knockdown expression of the specific denoted gene. The *C. elegans* strain, number of subjects, restricted mean (days), standard error, 95% confidence interval (CI), 95% median CI, and p-values for relevant comparisons are noted among all conditions. ns, not significant; *, p<0.05; **, p<0.01; ***, p<0.001; ****, p<0.0001 by log-rank analysis.

• MDAR checklist

## Data availability
All data generated or analyzed during this study are included in the manuscript and supporting files; Source Data files have been provided for Figure 2 as Figure 2 - source data 1. These same data have been made publicly available and can be found at Dryad.

The following dataset was generated:

| Author(s) | Year | Dataset title | Dataset URL | Database and Identifier |
|---|---|---|---|---|
| Cedillo L, Ahsan FM, Li S, Stuhr N, Zhou Y, Zhang Y, Adedoja A, Murphy LM, Yerevanian A, Emans S, Dao K, Li Z, Peterson ND, Watrous J, Jain M, Das S, Pukkila-Worley R, Curran SP, Soukas AA | 2023 | Ether Lipid Biosynthesis Promotes Lifespan Extension and Enables Diverse Prolongevity Paradigms in *Caenorhabditis elegans* | https://doi.org/10.5061/dryad.wdbrv15tj | Dryad Digital Repository, 10.5061/dryad.wdbrv15tj |

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
