## [Editor Report]

This paper explores the molecular basis underlying metformin treatment to understand why it is such an effective drug for improving age-related health and lifespan. Using *C. elegans* as a model organism in which to do this, the paper hones in on the role of ether lipid biosynthesis as an effector of metformin, and more broadly as a process implicated in extending lifespan in response to diet, TOR signalling and mitochondrial based interventions. The compelling data substantially support the conclusions and the better understanding of biguanide impact on metabolism is highly important in the field.

---

## [Decision Letter]

**Decision letter after peer review:**

Thank you for submitting your article "Ether Lipid Biosynthesis Promotes Lifespan Extension and Enables Diverse Prolongevity Paradigms in *Caenorhabditis elegans*" for consideration by *eLife*. Your article has been reviewed by 2 peer reviewers, and the evaluation has been overseen by a Reviewing Editor and Carlos Isales as the Senior Editor. The reviewers have opted to remain anonymous.

The reviewers and the reviewing editor find the paper important and compelling.

Essential revisions:

1) Experiments were done in the presence of FuDR. Given the potential side effect of this compound, it would be important to show that the key findings of ether lipid requirements for biguanide-mediated longevity are not affected by FuDR.

2) Further, it is important to assess the possibility that biguanides act via an effect on bacterial growth and thus indirect effects on worm metabolism.

3) Please explain in the text why different biguanide concentrations were used in different experiments.

4) Please address in the text the reviewer's question on the role of oxidative stress and the role of gst-4.

*Reviewer #1 (Recommendations for the authors):*

– Lipid metabolism and SKN-1 have been implicated in germline disruption longevity pathways (one example of several papers is PMID: 26196144). The longevity data here all appear to have been collected in the background of the fertility inhibitor FuDR. Although the use of FuDR is common in the field and justified, synergistic effects have been documented. The authors should address the question as to whether the key finding of ether lipid requirements for biguanide-mediated longevity is independent of the presence of FuDR.

– The authors state that biguanides do not stimulate expression of the skn-1 antioxidant defense effector gst-4 (Discussion, lines 580 – 582), and cite Cabreiro et al. 2013 and Onken and Driscoll, 2010 to support this – in fact, both these papers demonstrate the opposite, showing that biguanide treatment does result in increased gst-4 expression. This calls into question the author's suggestion that biguanides activate ether lipid biosynthesis and skn-1 signaling to trigger metabolic stress defenses rather than canonical oxidative stress pathways.

Minimally, this needs to be straightened out in terms of accurate citation and discussion.

Better, the argument might be addressed by looking at daf-16 and skn-1 downstream target expression with biguanide treatment or in fard-1(oe) mutants with and without ether lipid biosynthesis gene disruptions. Resistance to oxidative stress could be tested under similar conditions.

Less critical points that should be addressed:

The paper would benefit from just a little bit more description of the original genetic screen, although published in Wu 2016. The endpoint of the original screen was for growth/body size changes in high metformin; how did data translate to the longevity endpoint focus here?

Metformin is used in the clinic, phenformin is much less prescribed due to side effect complications. The authors do address reasons they chose to use phenformin in most studies, but the clinical differences between metformin and phenformin underscore how interesting it is to ask the question as to what the metabolic differences might be. A complete analysis is beyond this paper's scope, but simple testing for key findings, even simple lipid droplet impact and fat distribution using metformin (as in Figure 6) could define lipid distribution outcomes as a common feature; or possibly identify a difference.

One naturally wonders whether the over-expression of enzymes ads-1 and/or acl-7 would also be bioactive. I do not think these are essential studies to add, but if the authors have the information they should add it to the paper.

– Figure 1-B-D; E-G seems to share the same WT control for all presented data, authors should indicate that all tests run in the same control experiment if this is the case, and should confirm in methods that same-day WT controls are run in the same experimental test of each study.

– ads-1 and acl-7 mutants exhibit modest increases in longevity with metformin treatment; acl-7 + phen might be different at end stages. Authors do comment that the response is blunted but some note that not all impact of the drug is eliminated should be added.

– Although the text suggests fard-1, ads-1, acl-7 mutants are null or reduction of function, the assignment as to whether a particular locus is null, missense is not clear; please add a comment in methods.

In Figure 2 —figure supplement 2, the authors examine the localization of exogenously expressed FARD-1, and state that the localization of FARD-1 is not regulated by biguanide treatment (Results, lines 286 – 304). As FARD-1 is likely overexpressed in these experiments (as the authors themselves point out in later lifespan experiments using the fard-1 (oe1) construct, Figure 5A), it is possible that the localization pattern here is the result of an "activated" ether biosynthesis pathway state, and that further stimulation with biguanide treatment might not impact FARD-1 localization.

This caveat should be discussed; better for the authors to examine localization with FARD-1 expressed at endogenous levels with and without phenformin treatment.

The authors show that expression of ether lipid biosynthesis genes does not increase with biguanide treatment (Figure 2 —figure supplement 2; Results lines 305 – 313), and FARD-1::RFP levels decrease with phenformin treatment (Figure 2 —figure supplement 2M) and suggest that "post-translational negative feedback of ether lipids on the ether lipid biosynthetic pathway, as has been previously reported" (Results lines 311 – 313)--a citation for these results should be added.

These data argue against phenformin and fard-1 overexpression triggering downstream pathways in the same way. Although fard-1 oe increases lifespan in a manner that is not additive with phenformin treatment (Figure 5 A and B), fard-1 overexpression evidently does not trigger the same negative feedback mechanism, as overexpressed FARD-1 levels shown in Figure 2 —figure supplement 2 M appear to be high in vehicle controls, and are lowered with phenformin treatment. If both fard-1 overexpression and phenformin treatment signal similarly to trigger a post-translational negative feedback mechanism (which would presumably be important for lifespan extension), one would expect similar FARD-1 levels in fard-1 oe animals with and without phenformin treatment. It is possible that fard-1 overexpression simply overwhelms – or is insufficient to trigger – negative feedback mechanisms. Data showing overexpressed fard-1 mRNA and FARD-1 protein levels vs. endogenously expressed fard-1 and FARD-1, with and without biguanide treatment, would help clarify these seemingly paradoxical results.

Figure 3D (D-H) RNAi of three fatty acid desaturases (D-F) two fatty acid elongases (G and H) …blunt phenformin-mediated lifespan extension in wild type worms.

fat-1 RNAi extends lifespan, phen treatment brings this close to wt +phen, and there is no significant difference wt+phen vs. fat-1(RNAi) + phen; it might be technically reasonable to indicate the response is blunted (percent increase) but fat-1 does not appear to be all that impactful on longevity. These data do not impress regarding the importance of fat-1 in phenformin outcome; the discussion should be toned down or eliminated.

Page 439 Figure 6 legend Asdf in 3, independent ether..comma not needed.

In Figure 6, the authors show that phenformin inhibits fat accumulation in young adults in a manner that requires ether lipid biosynthesis gene expression. As a skn-1 gain-of-function mutant has a similar fat reduction phenotype that does not appear to be additive with phenformin treatment, and because lifespan extension with phenformin and lipid ether biosynthesis gene fard-1 overexpression requires skn-1, the authors suggest that biguanides increase ether lipid biosynthesis to trigger a low-fat, pro-longevity metabolic state through skn-1. To support this model, the authors could further explore the link between phenformin, ether lipid biosynthesis, and skn-1: is skn-1 expression required for the low-fat phenotype seen with phenformin treatment? Is the expression of ether lipid biosynthesis genes needed for the low-fat phenotype seen in skn-1 gf mutants? Do fard-1(oe) mutants display a similar low-fat phenotype? What is the state of ether lipid levels in skn-1 gf mutants (as compared to phenformin-treated animals in Figure 2E and fard-1 overexpressors in Figure 5G)?

Figure 5C-text suggests all three kds disrupt, not just ads-1, why not include data for other kds in the main figures?

Figure 7B model: Dietary restriction executed is really one type of a complex response, so it is a bit of an over-extension to extend to DR in general; this can be addressed by indicating eat-2 DR in the box. Comments on DR would be more compelling if a second model of DR-associated longevity were tested.

daf-16 is not needed for biguanide-associated longevity, but it is for fard-1(oe)-mediated longevity -either explain why the dashed lines are used for daf-16 in the legend or better adjust the figure to visually reflect this complexity.

Discussion line 499 start. Our results show that the biguanides metformin and phenformin promote lifespan extension by stimulating the biogenesis of ether lipids, prompting longevity promoting, metabolic stress defenses mediated by skn-1.

Data relate to phenformin action but not necessarily metformin. Please adjust.

Possible point for additional discussion. Metformin has been documented to intersect with the bacterial food source to influence longevity PMID: 23540700. This manuscript does not engage that story, but I would encourage authors to add a small statement regarding their take on the interface of the "microbiome" or bacterial influence on biguanide action and possibly ether lipid synthesis.

Finally, as the authors point out, identification of the specific lipids necessary for promoting healthy aging is not feasible here; with which I totally agree. The actual mechanism by which ether lipid synthesizing enzymes modulate extended survival for the isp-1, eat-2, and raga-1 backgrounds or +phenformin is not really defined by this work. Still, the highlight of a critical biosynthetic pathway in biguanide outcomes and the implication of SKN-1/NRF2 in the biology is of sufficient importance in the field to merit high-profile publication.

*Reviewer #2 (Recommendations for the authors):*

Key points to address:

– Address the impact of biguanides on bacterial growth (OP50, HT115) experimentally e.g. bacterial growth, the impact of heat-killed bacteria on key biguanide phenotypes, and their relationship to ether lipids. The impact of biguanide concentration should also be considered in this regard.

– Address whether different amounts of biguanide cause the same effect on ether lipids.

– Address the direct genetic connections between skn-1 and fard-1 (etc) e.g. by creating a fard-1; skn-1 gof double and examining fat +/- Phenformin. Alternatively, could this be examined using fat measurements from aak-2 and daf-16 +/- fard-1? It would be interesting to see if the mechanism is the same for all of these longevity processes and make the study of even broader relevance.

Other points:

– Are the error bars in Figure 1 S1A correct? The data is normalised to 1. I am also unclear of the logic here. Metformin makes the worms smaller, but fard-1 and acl-7 RNAi reverse this? Is this the same with the mutants?

– Line 283 – 'the' missing? + other punctuation errors scattered about.

– The theory that the web-like structures marked by FARD-1::RFP are the ER is intriguing. What magnification are these images taken at in Figure 2 S2? Readers would need that information to decipher this.

– The authors mention that a pathway of negative feedback on ether lipid biosynthesis has been reported please could you supply that reference as it is currently missing – line 313.

– How are the numbers of lipid droplets measured in Figure 6? Per area? Per whole worm?

– Please address the issue of skn-1 isoforms in the text. You have carried out skn-1 RNAi which in theory takes out all isoforms – have you confirmed this? My issue is whether the skn-1 gof mutants and the skn-1 RNAi are true opposites of each other. It is also important as you refer to it as skn-1/Nrf2 specifically. You could just remove the 2.

---

## [Author Response]

Essential revisions:1) Experiments were done in the presence of FuDR. Given the potential side effect of this compound, it would be important to show that the key findings of ether lipid requirements for biguanide-mediated longevity are not affected by FuDR.

We thank the reviewers for this important comment, and now include as Figure 1 —figure supplement 2 lifespan analyses that rigorously indicate that loss of function mutations in the ether lipid biosynthetic machinery completely blunt both metformin and phenformin-mediated lifespan extension without the use of FUdR, with results comparable to that of FUdR -treated animals as previously shown. We include discussion of these results in the manuscript between lines 160 and 165.

2) Further, it is important to assess the possibility that biguanides act via an effect on bacterial growth and thus indirect effects on worm metabolism.

We thank the reviewers for highlighting this potential caveat. To address whether biguanide-mediated ether lipid induction relies on live bacteria, we leveraged a robust strategy to prevent both bacterial replication and metabolic activity of the OP50-1 *E. coli* food source via pre-treatment with 1% paraformaldehyde (PFA) for two hours prior to seeding on nematode growth medium (NGM) agar plates^3^. We confirm that 1% PFA treatment completely prevents bacterial replication, and that the pre-treatment process itself does not significantly abrogate bacterial titer production compared to standard OP50-1 plate seeding protocols (Figure 6 —figure supplement 2A). Using this approach, we verify through lipidomic FAME GC/MS analysis that biguanides drive somatic fatty acid depletion and increases 16:0 DMA species. Importantly, these effects do not rely on the OP50-1 food source being alive, suggesting that the effects of biguanides on increasing ether lipid biosynthesis and altering somatic lipid stores do not act directly through alterations in bacterial growth or metabolism (Figure 6 —figure supplement 2B-G). We additionally include lifespan analyses that reveal that biguanides still extend lifespan on metabolically killed OP50-1, and that ether lipid deficiency in *ads-1* mutant animals completely abrogates both metformin and phenformin-mediated lifespan extension, even when grown on metabolically inactivated and dead bacteria (Figure 6 —figure supplement 3). Combined, these data rigorously advance the conclusion that the effects of biguanides on ether lipid biosynthesis and lifespan extension occurs directly through the nematode, and do not rely upon alterations in bacterial growth or metabolism to exert their pro-longevity effects. We include discussion of these new results in lines 509 to 547. Given that these results contrast with previously published work by others, and the importance to the field, we plan a whole separate manuscript about biguanide effects directly on nematode lifespan versus indirectly through the bacteria. As such, we do not dwell extensively on the subject herein.

3) Please explain in the text why different biguanide concentrations were used in different experiments.

We thank the reviewers for the opportunity to clarify the uses of different concentrations of biguanides in our assays. We wish to clarify that our current study is focused on the role of ether lipid machinery in modulation of lifespan and healthy aging – as such, we focused on genetic and biochemical experiments leveraging minimally lifespan extending doses of biguanides (4.5 mM phenformin and 50 mM metformin) to identify epistatic links and interactions, as utilized by several independent laboratories ^4-9^. The goal of the 160 mM dosage of metformin used in our prior genetic screens ^10^ and subsequently highlighted in Figure 1 —figure supplement 1A is to enhance the sensitivity and specificity of our discovery approach to identify effectors of the biological action of biguanides. The 160 mM dose we leveraged causes potent growth inhibition ^10^. Our prior published work indicates that using this dose to look for gene inactivations that block the growth inhibitory effects of the drug can also identify longevity effectors of metformin^10^. Thus, we used a similar strategy here to identify *fard-1* and *acl-7,* which were initially identified as gene knockdowns that block the growth inhibitory effects of 160 mM metformin. Thereafter, lower doses of biguanides are used (4.5 mM phenformin and 50 mM metformin) to elucidate requirements of the ether lipid machinery in biguanide-prompted longevity. We succinctly modify in the text the reasoning for the different biguanide concentrations used in this work (lines 135 to 153).

4) Please address in the text the reviewer's question on the role of oxidative stress and the role of gst-4.

We thank the reviewers for allowing us to highlight this important clarification. We have performed reporter imaging experiments in the *gst-4* transcriptional reporter strain *gst-4p::GFP::NLS* (included as Figure 6 —figure supplement 1A-B), suggesting that biguanides do not significantly activate *gst-4* expression in either OP50 and HT115 dietary sources, and that phenformin treatment conversely significantly reduced *gst-4* expression. We have repeated this experiment numerous times and at multiple timepoints in the lifespan. In fact, the analysis of *gst-4* expression indicates that metformin and phenformin ***decrease** gst-4* expression. These data substantiate our claim that the requirement of SKN-1 activity downstream of biguanides to mediate somatic lipid depletion and pro-longevity outcomes is not due to a *gst-4* mediated antioxidant role. We additionally include reporter imaging data for expression of *dod-24*, a metabolic defense stress response factor known to be transcriptionally regulated by SKN-1^2^, revealing that biguanides activate *dod-24* expression via a mechanism dependent on both ether lipid machinery and SKN-1 activity (Figure 6F). Concordant with our prior work^2^, there are multiple “flavors” of *skn-1* activation, and the metabolic “flavor” is oppositely regulated to oxidative defenses (thus leading to increased expression of *dod-24* and decreased expression of *gst-4*). Given that these results conflict with the literature and the prevailing notions in the field. we describe these results in the revised text in the Results section in lines 497 to 508 and in the Discussion in lines 659 to 673.

Reviewer #1 (Recommendations for the authors):1. Lipid metabolism and SKN-1 have been implicated in germline disruption longevity pathways (one example of several papers is PMID: 26196144). The longevity data here all appear to have been collected in the background of the fertility inhibitor FuDR. Although the use of FuDR is common in the field and justified, synergistic effects have been documented. The authors should address the question as to whether the key finding of ether lipid requirements for biguanide-mediated longevity is independent of the presence of FuDR.

We thank the reviewer for this suggestion. As noted in Author Response 1, we have performed lifespan experiments without the use of FUdR (included as Figure 1 —figure supplement 2) for wild type and ether lipid deficient mutants treated with either metformin or phenformin in biological triplicate. These new data reveal that the requirements for the ether lipid machinery in biguanide-mediated lifespan extension are independent of the effects of FUdR, with abrogation of lifespan extension seen in rates comparable to that when using FUdR (Figure 1B-G and Supplementary file 1). We address these important confirmatory results in the text under lines 160 to 165.

2. The authors state that biguanides do not stimulate expression of the skn-1 antioxidant defense effector gst-4 (Discussion, lines 580 – 582), and cite Cabreiro et al. 2013 and Onken and Driscoll, 2010 to support this – in fact, both these papers demonstrate the opposite, showing that biguanide treatment does result in increased gst-4 expression. This calls into question the author's suggestion that biguanides activate ether lipid biosynthesis and skn-1 signaling to trigger metabolic stress defenses rather than canonical oxidative stress pathways.Minimally, this needs to be straightened out in terms of accurate citation and discussion.Better, the argument might be addressed by looking at daf-16 and skn-1 downstream target expression with biguanide treatment or in fard-1(oe) mutants with and without ether lipid biosynthesis gene disruptions. Resistance to oxidative stress could be tested under similar conditions.

We thank the reviewer for highlighting this textual oversight, and for the opportunity to better expand upon our suggestion that biguanides rely upon SKN-1 induced metabolic stress defense to exert a pro-longevity outcome, independent of its known canonical oxidative stress response functions. We have corrected the discussion of the Onken and Driscoll, 2010 and Cabreiro et al., 2013 citations to highlight the previously published observations of a moderate ~1.5 to 2-fold increase in *gst-4* expression with 50 mM metformin treatment^4,5^. We have attempted numerous times to reproduce this induction result with both 50 mM metformin and 4.5 mM phenformin administration, at multiple times across the lifespan. We include as Figure 6 —figure supplement 1A-B data that fails to replicate the subtle increase in *gst-4* expression previously reported on either OP50-1 or HT115 food sources, and instead show that phenformin treatment significantly reduces *gst-4* expression. We include as Figure 6F data that indicate that phenformin treatment activates expression of the metabolic stress defense factor *dod-24*, a known transcriptional target of the non-canonical SKN-1 innate immune effector regulon^2^. The induction of *dod-24p::GFP* expression is both *skn-1* and ether lipid machinery dependent, analogous to the dependencies seen with adult somatic lipid depletion (Figure 6B-E). Thus, we argue that biguanides exert their pro-longevity outcomes via alterations in somatic lipid levels and metabolic stress defense response through an ether lipid and SKN-1 signaling axis independent of the transcription factor’s known role in inducing an oxidative stress response. We have included discussion of these new results in Results (lines 497-508) and (659-673) of the manuscript. The decrease in *gst-4* expression seen with phenformin treatment is potentially of great interest, suggesting that phenformin treatment may increase the sensitivity of animals to oxidative stressors. We are focusing on follow ups of this result in manuscripts currently in preparation.

Less critical points that should be addressed:3. The paper would benefit from just a little bit more description of the original genetic screen, although published in Wu 2016. The endpoint of the original screen was for growth/body size changes in high metformin; how did data translate to the longevity endpoint focus here?

We thank the reviewer for the opportunity to highlight the rationale for our original genetic screen in integration of growth and longevity outcomes. In our original study, we leveraged an RNAi library consisting of ~1000 genes annotated with a metabolic gene ontology to identify genetic elements required for metformin-mediated growth inhibition, to illuminate the physiologic mechanisms that regulate the drug’s known anti-neoplastic and anti-growth proliferative effects. We identified a pathway linking mitochondrial perturbation to improved nuclear pore complex (NPC) and nucleocytoplasmic trafficking function, resulting in reduced mTORC1 signaling and activation of biguanide response element CeACAD10 to induce growth inhibition^10^. We surprisingly found that improved NPC fidelity is not only required for the growth inhibitory properties of the drug, but were also necessary for the pro-longevity outcomes, suggesting a unified mechanism for both anti-cancer and lifespan extension^10^. Given our previous success in identifying genetic elements that bridge both the growth inhibitory and pro-longevity mechanisms of the drug, we were interested in identifying other hits from our genetic screen that may be required for both mechanisms, leading to our current study interrogating the role of ether lipid machinery in regulation of biguanide-mediated lifespan extension. We have amended the text in lines 135 to 153 to succinctly describe this transition.

4. Metformin is used in the clinic, phenformin is much less prescribed due to side effect complications. The authors do address reasons they chose to use phenformin in most studies, but the clinical differences between metformin and phenformin underscore how interesting it is to ask the question as to what the metabolic differences might be. A complete analysis is beyond this paper's scope, but simple testing for key findings, even simple lipid droplet impact and fat distribution using metformin (as in Figure 6) could define lipid distribution outcomes as a common feature; or possibly identify a difference.

We agree with the reviewer that the potential link between clinical differences and metabolic outcomes of metformin and phenformin administration are alluring to investigate and of upmost importance to clarify. We are currently preparing a manuscript in which we thoroughly dissect the differential metabolic responses to metformin and phenformin in nematodes, and highlight their relationship to stress response, pro-longevity, and growth inhibitory outcomes. We believe that the lipid distribution experiments as suggested by the reviewer will be of critical importance in our next manuscript but is beyond the scope of this manuscript, specifically highlighting the unified importance of ether lipid machinery in both metformin and phenformin-mediated lifespan extension.

5. One naturally wonders whether the over-expression of enzymes ads-1 and/or acl-7 would also be bioactive. I do not think these are essential studies to add, but if the authors have the information they should add it to the paper.

We thank the reviewer for this insightful suggestion. We are currently working on developing these genetic reagents for future studies that will finely delineate the specific nodes during ether lipid biosynthesis that may confer pro-longevity and healthspan benefits in the nematode.

6. Figure 1-B-D; E-G seems to share the same WT control for all presented data, authors should indicate that all tests run in the same control experiment if this is the case, and should confirm in methods that same-day WT controls are run in the same experimental test of each study.

We apologize for the oversight in clearly highlighting that the lifespan analyses shown in Figure 1B-D;E-G are representative of results from 1 replicate run in the same batch with same-day wildtype controls. We amend in the figure legends for Figure 1 and Materials and methods sections that this is the case (lines 179 to 180) and note that additional replicates with independent wild type controls and all appropriate statistics for comparisons are included as Supplementary file 1.

7. ads-1 and acl-7 mutants exhibit modest increases in longevity with metformin treatment; acl-7 + phen might be different at end stages. Authors do comment that the response is blunted but some note that not all impact of the drug is eliminated should be added.

We thank the reviewer for their careful interpretation of our lifespan analyses. We have amended the text in lines 153 to 155 to note that *a**ds-1* and *a**cl-7* mutants may display a modest increase in lifespan with metformin administration, but with a percentage median lifespan increase significantly reduced in comparison to wildtype controls (Supplementary file 1). We also note that although these mutations result in a loss of function of the protein, they are missense mutations, and thus a small amount of residual ether lipid biosynthetic capacity cannot be completely ruled out (see the following comment in the response to point 8 below).

8. Although the text suggests fard-1, ads-1, acl-7 mutants are null or reduction of function, the assignment as to whether a particular locus is null, missense is not clear; please add a comment in methods.

We have included in the Strain subsection in the Materials and methods (lines 721 to 723) a description of the genetic lesions in each of the mutants. *fard-1(wa28)* (G261D), *acl-7 (wa20)* (R234C), and *ads-1(wa3)* (G454D) are all missense mutations that result in a loss of nearly all detectable DMA synthesis^11^. Although these mutations generate a loss-of-function effect, a small amount of residual ether lipid biosynthetic capacity cannot be completely ruled out in these animals.

9. In Figure 2 —figure supplement 2, the authors examine the localization of exogenously expressed FARD-1, and state that the localization of FARD-1 is not regulated by biguanide treatment (Results, lines 286 – 304). As FARD-1 is likely overexpressed in these experiments (as the authors themselves point out in later lifespan experiments using the fard-1 (oe1) construct, Figure 5A), it is possible that the localization pattern here is the result of an "activated" ether biosynthesis pathway state, and that further stimulation with biguanide treatment might not impact FARD-1 localization.10. This caveat should be discussed; better for the authors to examine localization with FARD-1 expressed at endogenous levels with and without phenformin treatment.

We appreciate the reviewer highlighting the important caveat of relying on exogenous overexpression systems for colocalization and co-interaction studies. We include this as a limitation of our current study in lines 650 to 658, and efforts are currently underway to endogenously tag FARD-1 but are not available at the time of this revision.

11. The authors show that expression of ether lipid biosynthesis genes does not increase with biguanide treatment (Figure 2 —figure supplement 2; Results lines 305 – 313), and FARD-1::RFP levels decrease with phenformin treatment (Figure 2 —figure supplement 2M) and suggest that "post-translational negative feedback of ether lipids on the ether lipid biosynthetic pathway, as has been previously reported" (Results lines 311 – 313)--a citation for these results should be added.

We apologize for this oversight and have included the citation for Honsho, et al. 2010 in lines 312 and 648^1^.

12. These data argue against phenformin and fard-1 overexpression triggering downstream pathways in the same way. Although fard-1 oe increases lifespan in a manner that is not additive with phenformin treatment (Figure 5 A and B), fard-1 overexpression evidently does not trigger the same negative feedback mechanism, as overexpressed FARD-1 levels shown in Figure 2 —figure supplement 2 M appear to be high in vehicle controls, and are lowered with phenformin treatment. If both fard-1 overexpression and phenformin treatment signal similarly to trigger a post-translational negative feedback mechanism (which would presumably be important for lifespan extension), one would expect similar FARD-1 levels in fard-1 oe animals with and without phenformin treatment. It is possible that fard-1 overexpression simply overwhelms – or is insufficient to trigger – negative feedback mechanisms. Data showing overexpressed fard-1 mRNA and FARD-1 protein levels vs. endogenously expressed fard-1 and FARD-1, with and without biguanide treatment, would help clarify these seemingly paradoxical results.

We thank the reviewer for this important suggestion, and we have performed the experiment as suggested. We include in Figure 5G-H qRT-PCR analysis of wild type and *fard-1(oe3)* animals treated with phenformin, evaluating the RNA expression levels of endogenous *fard-1* (using qRT-PCR primers probing for the native 3’ UTR of *fard-1*, Figure 5D). These results indicate that FARD-1 overexpression indeed reduces native *fard-1* expression by ~40%, and that biguanide treatment does not additively reduce *fard-1* expression even further. Interestingly, biguanide treatment appears to reduce the expression of exogenously overexpressed *fard-1* by about ~60%, consistent with the reduction in RFP-tagged protein levels shown in Figure 2 —figure supplement 2M. We speculate that the discrepancy in feedback mechanisms between the endogenous and exogenous *fard-1* transcripts is likely due to the design of the overexpressor, as the *fard-1(oe)* strains in this manuscript all utilize a synthetic *unc-54* 3’ UTR to enhance permissive expression in all *C. elegans* somatic cells, and likely maintain expression as a multicopy integrant in the genome, thus overwhelming negative feedback mechanisms. Indeed, several studies have suggested that the 3’ UTR structure of several mRNAs mediate negative feedback loops to finely tune protein expression patterns^12,13^. Combined, these results suggest that FARD-1 overexpression and biguanide treatment regulate endogenous *fard-1* levels through similar mechanisms, but the differential UTR utilization and potential for multicopy integration may explain their distinct lifespan promoting dependencies. We discuss the significance of these results in lines 650 through 655.

13. Figure 3D (D-H) RNAi of three fatty acid desaturases (D-F) two fatty acid elongases (G and H) …blunt phenformin-mediated lifespan extension in wild type worms.fat-1 RNAi extends lifespan, phen treatment brings this close to wt +phen, and there is no significant difference wt+phen vs. fat-1(RNAi) + phen; it might be technically reasonable to indicate the response is blunted (percent increase) but fat-1 does not appear to be all that impactful on longevity. These data do not impress regarding the importance of fat-1 in phenformin outcome; the discussion should be toned down or eliminated.

We thank the reviewer for their careful interpretation of the lifespan results for *fat-1* RNAi. We agree that there does not appear to be a meaningful or interpretable interaction between *fat-1* and biguanide action in longevity, and so we have removed the *fat-1* data and reference to it from this version of the text.

14. Page 439 Figure 6 legend Asdf in 3, independent ether..comma not needed

We have deleted this comma and thoroughly checked the remainder of the manuscript for punctuation and grammatical errors.

15. In Figure 6, the authors show that phenformin inhibits fat accumulation in young adults in a manner that requires ether lipid biosynthesis gene expression. As a skn-1 gain-of-function mutant has a similar fat reduction phenotype that does not appear to be additive with phenformin treatment, and because lifespan extension with phenformin and lipid ether biosynthesis gene fard-1 overexpression requires skn-1, the authors suggest that biguanides increase ether lipid biosynthesis to trigger a low-fat, pro-longevity metabolic state through skn-1. To support this model, the authors could further explore the link between phenformin, ether lipid biosynthesis, and skn-1: is skn-1 expression required for the low-fat phenotype seen with phenformin treatment? Is the expression of ether lipid biosynthesis genes needed for the low-fat phenotype seen in skn-1 gf mutants? Do fard-1(oe) mutants display a similar low-fat phenotype? What is the state of ether lipid levels in skn-1 gf mutants (as compared to phenformin-treated animals in Figure 2E and fard-1 overexpressors in Figure 5G)?

We thank the reviewer for these highly insightful suggestions to better delineate the link between SKN-1 and ether lipid synthesis with biguanide-mediated low-fat metabolic states. We now include as Figure 6D-E Asdf analysis of wildtype and *skn-1(zu135)* complete loss-of-function animals treated with phenformin, showing that SKN-1 is absolutely required for biguanide-mediated somatic depletion of fat. We amended Figure 6B-C to include Asdf analysis of FARD-1 overexpressing animals, indicating that *fard-1 (oe1)* animals indeed constitutively activate a low somatic fat state. To elucidate the state of ether lipid levels in *skn-1gf* mutants, we performed FAME GC/MS analysis of lipids extracted from *skn-1gf(lax188*) animals relative to wildtype (N2) at adult day 2 of lifespan. As shown in Author response image 1, *skn-1gf* animals differentially alter ether lipid synthesis in a manner distinct from FARD-1 overexpression indicated in Figure 5I, increasing 18:0 DMA levels while nominally altering 16:0 DMA and 18:1 DMA levels. Combined, these three results corroborate the model that biguanides induce a somatic fat depleted state through a mechanism dependent on SKN-1 activity downstream of activation of the ether lipid biosynthetic machinery.

**Author response image 1. sa2fig1:** Gain of function mutation in SKN-1 results in global depletion of free fatty acid levels and elevates ether lipid precursor alcohols. (A) Quantification of total area under the curve (AUC) measurements for all identified free fatty acids (FFA) in wildtype (N2) and *skn-1gf(lax188)* animals at Adult Day 2 using fatty acid methyl ester extraction followed by gas chromatography – mass spectrometry (FAME GC/MS). (B) Quantification of alkenyl dimethylacetal fatty alcohols (DMA) in N2 and *skn-1gf(lax188)* animals at Adult Day 2 using FAME GC/MS. Data are total sum normalized to percentage of total FFA pool. For A-B, data represent the mean +/- SEM of n = 4 independent biological replicates. *, P < 0.05; **, P < 0.01; ***, P < 0.001; ****, P < 0.0001 by Student’s t-test (A) or multiple t-tests with multiple hypothesis correction by two-stage step-up method of Benjamini, Krieger, and Yekutieli (B).

16. Figure 5C-text suggests all three kds disrupt, not just ads-1, why not include data for other kds in the main figures?

We appreciate the reviewers’ suggestion, and, in addition to *ad*s-1 (Figure 5C) also now include data indicating that RNAi knockdown of *fard-1* and *a*cl-7 similarly suppresses FARD-1 overexpression -mediated lifespan extension as Figure 5 —figure supplement 1A-B.

17. Figure 7B model: Dietary restriction executed is really one type of a complex response, so it is a bit of an over-extension to extend to DR in general; this can be addressed by indicating eat-2 DR in the box. Comments on DR would be more compelling if a second model of DR-associated longevity were tested.

We appreciate the reviewers’ comments and have amended the model in Figure 7B to now state ‘EAT-2 DR’.

18. daf-16 is not needed for biguanide-associated longevity, but it is for fard-1(oe)-mediated longevity -either explain why the dashed lines are used for daf-16 in the legend or better adjust the figure to visually reflect this complexity.

We now explicitly detail the reasoning for the dashed lines in DAF-16 in the legend for Figure 7B (lines 713 to 716). We wished to indicate the point that the reviewer suggests – that DAF-16 is required for FARD-1 overexpression but not biguanide-mediated lifespan extension.

19. Discussion line 499 start. Our results show that the biguanides metformin and phenformin promote lifespan extension by stimulating the biogenesis of ether lipids, prompting longevity promoting, metabolic stress defenses mediated by skn-1.Data relate to phenformin action but not necessarily metformin. Please adjust.

We adjust the text as suggested by the reviewer to indicate phenformin only in line 577.

20. Possible point for additional discussion. Metformin has been documented to intersect with the bacterial food source to influence longevity PMID: 23540700. This manuscript does not engage that story, but I would encourage authors to add a small statement regarding their take on the interface of the "microbiome" or bacterial influence on biguanide action and possibly ether lipid synthesis.

We thank the reviewer for this critical suggestion, which was additionally suggested by Reviewer #2 and the Reviewing Editor for experimental evaluation. As indicated in Author Response 3, we have performed FAME GC/MS of Adult Day 1 nematodes treated with or without phenformin and grown on live or metabolically dead OP50-1 *E. coli* food sources^3^. These analyses rigorously show that biguanides increase nematode ether lipid levels independent of any indirect effects on bacterial growth, proliferation, or metabolism (Figure 2 —figure supplement 3). Compellingly, metabolically inactivating the bacterial food source does not impair the ability of the ether lipid machinery to abrogate biguanide-mediated lifespan extension (Figure 2 —figure supplement 4). Additionally, our previously published study interrogating the role of CeACAD10 in regulation of metformin action in growth and cancer highlights that UV-mediated inactivation of OP50-1 growth does not affect metformin-mediated growth inhibition, and that direct injection of metformin into the animal is sufficient to drive CeACAD10 biguanide response activation^14^. Combined, these data rigorously exclude the possibility that biguanides alter ether lipid levels or the activity of the machinery via indirect effects through bacterial metabolism. We include a new section in Results to highlight this important statement (lines 509 to 547) and include a brief description of these results in the Discussion (lines 577 to 580).

21. Finally, as the authors point out, identification of the specific lipids necessary for promoting healthy aging is not feasible here; with which I totally agree. The actual mechanism by which ether lipid synthesizing enzymes modulate extended survival for the isp-1, eat-2, and raga-1 backgrounds or +phenformin is not really defined by this work. Still, the highlight of a critical biosynthetic pathway in biguanide outcomes and the implication of SKN-1/NRF2 in the biology is of sufficient importance in the field to merit high-profile publication.

We again thank the reviewer for their support of our work and for their excellent feedback, which greatly strengthened the mechanistic underpinnings of our work.

Reviewer #2 (Recommendations for the authors):Key points to address:1. Address the impact of biguanides on bacterial growth (OP50, HT115) experimentally e.g. bacterial growth, the impact of heat-killed bacteria on key biguanide phenotypes, and their relationship to ether lipids. The impact of biguanide concentration should also be considered in this regard.

We thank the reviewer for this important suggestion, and we have performed the experiment as suggested. As indicated in Author Responses 3 and 22, we have performed FAME GC/MS of adult day 1 nematodes treated with or without phenformin and grown on live or metabolically dead OP50-1 *E. coli* food sources using a rigorously established 1% PFA treatment protocol ^3^. This PFA treatment protocol, as opposed to heat or UV killing protocols, both completely kill and metabolically inactivate the food source (Figure 6 —figure supplement 2A), making this a more reliable, uniformly consistent strategy to eliminate bacterial growth and metabolism as confounding variables in lipidomic and lifespan analyses ^3^. These analyses rigorously show that biguanides increase relative nematode ether lipid levels independent of any indirect effects on bacterial growth, proliferation, or metabolism (Figure 6 —figure supplement 2D-G). Compellingly, metabolic inactivation of the bacterial food source neither impairs the lifespan extending effects of biguanides nor impairs the ability of the ether lipid machinery to abrogate biguanide-mediated lifespan extension (Figure 6 —figure supplement 3A-F). Combined, these data rigorously exclude the possibility that biguanides alter ether lipid levels or the activity of the machinery via indirect effects through bacterial growth or metabolism. We performed experiments utilizing a final dosage of 4.5 mM phenformin and 50 mM metformin, both of which are minimally required to robustly and reproducibly extend *C. elegans* lifespan, as corroborated and utilized by several independent laboratories^7-10^. Thus, these data strengthen the conclusion that phenformin and metformin prompt ether lipid-dependent increases in lifespan via direct effects on the worm, i.e. independently of biguanide effects on bacteria. Given that these results contrast with previously published work by others, and the importance to the field, we plan a whole separate manuscript about biguanide effects directly on nematode lifespan versus indirectly through the bacteria. As such we do not dwell extensively on the subject herein.

2. Address whether different amounts of biguanide cause the same effect on ether lipids.

We thank the reviewer for this suggestion. We again wish to clarify to the reviewer that the intention of this study is to illuminate the requirements for ether lipid biosynthesis on the pro-longevity mechanism of biguanides. As such, we utilized doses of metformin and phenformin that minimally robustly extend *C. elegans* lifespan (4.5 mM phenformin and 50 mM metformin) without resulting in negative pleiotropies, as utilized by multiple independent laboratories to identify genetic regulators of the drug^7-10^. The supraphysiological 160 mM dosage of metformin used in our prior genetic screens was only initially utilized to identify biguanide response elements. Given that the prior literature shows that higher doses of biguanides do not positively influence *C. elegans* lifespan^5,7^, even if additive effects on ether lipids were evident, we argue that they are not relevant to the major thrust of this manuscript on longevity-promoting effects of biguanides and ether lipids.

3. Address the direct genetic connections between skn-1 and fard-1 (etc) e.g. by creating a fard-1; skn-1 gof double and examining fat +/- Phenformin. Alternatively, could this be examined using fat measurements from aak-2 and daf-16 +/- fard-1? It would be interesting to see if the mechanism is the same for all of these longevity processes and make the study of even broader relevance.

We thank the reviewer for this suggestion. To clarify, we believe that SKN-1 is the transcription factor operating downstream of the ether lipid machinery to induce somatic lipid depletion, metabolic stress defenses, and pro-longevity outcomes. We have multiple lines of evidence to support this conclusion: (1) A *skn-1* RNAi targeting all isoforms of SKN-1 completely abrogates the pro-longevity effects of *fard-1 (oe1)* animals (Figure 5D), (2) a total SKN-1 loss-of-function mutant completely prevents both biguanide-mediated lifespan extension^4^ and somatic lipid depletion phenotypes (Figure 6D-E), and (3) *skn-1gf* animals displaying depleted somatic lipid content elevate 18:0 DMA levels (Response Figure 1), in contrast to changes in 16:0 DMA and 18:1 DMA in *fard-1 (oe3)* animals, suggesting that *skn-1gf* alone does not mirror FARD-1 overexpression mediated changes in ether lipid levels. As such, we do not believe that a *fard-1lf,skn-1gf* animal will suppress the *skn-1gf* Asdf phenotype. We agree that it would be interesting to see if multiple pro-longevity paradigms known to activate the major pro-longevity signaling and transcription factors AAK-2 and DAF-16 similarly control somatic lipid depletion, but we believe that this is outside of the scope of this initial manuscript delineating the role of SKN-1 and ether lipid machinery in regulation of biguanide-mediated lipid depletion and lifespan extension.

Other points:4. Are the error bars in Figure 1 S1A correct? The data is normalised to 1. I am also unclear of the logic here. Metformin makes the worms smaller, but fard-1 and aCl^-^7 RNAi reverse this? Is this the same with the mutants?

We thank the reviewer for the opportunity to clarify the results and significance of our growth inhibition assay in Figure 1 —figure supplement 1A. We have clarified in the figure legend that the bars in Figure 1 —figure supplement 1A are in fact SEM. The reviewer is correct in their interpretation of our data, that RNAi inactivation of *fard-1* and *acl-7* significantly blunt the growth inhibitory properties of the drug. These results corroborate the findings from our initial ~1000 metabolism gene RNAi screen for genetic elements that control metformin-induced growth inhibition, implicating the ether lipid machinery as critical for the growth inhibitory properties of the drug. Since this experiment was only used to identify potential response elements, we did not test the genetic mutants in the growth inhibition assay. Instead, we shifted our experimentation to address whether the ether lipid machinery is required for biguanide-mediated lifespan extension, which is the major thrust of the manuscript.

5. Line 283 – 'the' missing? + other punctuation errors scattered about.

We thank the reviewer for catching this typo. All authors have carefully examined the rest of this revised manuscript for punctuation and grammatical mistakes.

6. The theory that the web-like structures marked by FARD-1::RFP are the ER is intriguing. What magnification are these images taken at in Figure 2 S2? Readers would need that information to decipher this.

The images in Figure 2 —figure supplement 2 were taken using a Zeiss Plan-Apochromat 63x/1.4 Oil DIC M27 objective with a 2.0 scan zoom for each field. We have included the details of this magnification in the figure legend for Figure 2 —figure supplement 2F (lines 1463 to 1465)

7. The authors mention that a pathway of negative feedback on ether lipid biosynthesis has been reported please could you supply that reference as it is currently missing – line 313.

We apologize to the reviewer for our oversight on including this reference. We have included the appropriate reference^1^ as requested in lines 312 and 648.

8. How are the numbers of lipid droplets measured in Figure 6? Per area? Per whole worm?

We thank the reviewer for this important clarification. We include details regarding the measurements of lipid droplets in the Materials and methods section under ‘Lipid droplet analysis’ (lines 964 to 986). Z-stacked images were obtained for the intestine near the tail end of 14 *glo-4* RNAi, vehicle treated and 19 *glo-4* RNAi, phenformin treated worms (2 biological replicates per condition). 5 planes were extracted (planes 1,2,4,5, and 9) using ImageJ for all samples. For lipid droplet counting, quantification was performed using CellProfiler 4.2.1^15^ where lipid droplets were identified as primary objects. The min/max range for typical object diameters was 3-67 pixels, and those objects outside of the diameter range were discarded. Planes were excluded entirely if the pipeline did not accurately capture individual lipid droplets for most objects.

9. Please address the issue of skn-1 isoforms in the text. You have carried out skn-1 RNAi which in theory takes out all isoforms – have you confirmed this? My issue is whether the skn-1 gof mutants and the skn-1 RNAi are true opposites of each other. It is also important as you refer to it as skn-1/Nrf2 specifically. You could just remove the 2.

As suggested by the reviewer, we have changed all designations of *skn-1/Nrf2* to *skn-1/Nrf* throughout this manuscript. Although the sequence verified Ahringer Library *skn-1* RNAi clone utilized for RNAi knockdown experiments has been previously shown to target and abrogate expression of all 4 SKN-1 transcript isoforms^2,16-18^, we agree that we cannot rigorously conclude which SKN-1 isoform(s) is/are required for biguanide- or FARD-1 overexpression-mediated lifespan extension, and further, which orthologous Nrf function is represented.

References:

1. Honsho, M., Asaoku, S., and Fujiki, Y. (2010). Posttranslational regulation of fatty acyl-CoA reductase 1, Far1, controls ether glycerophospholipid synthesis. J Biol Chem *285*, 8537-8542. 10.1074/jbc.M109.083311.

2. Nhan, J.D., Turner, C.D., Anderson, S.M., Yen, C.A., Dalton, H.M., Cheesman, H.K., Ruter, D.L., Uma Naresh, N., Haynes, C.M., Soukas, A.A., et al. (2019). Redirection of SKN-1 abates the negative metabolic outcomes of a perceived pathogen infection. Proc Natl Acad Sci U S A *116*, 22322-22330. 10.1073/pnas.1909666116.

3. Beydoun, S., Choi, H.S., Dela-Cruz, G., Kruempel, J., Huang, S., Bazopoulou, D., Miller, H.A., Schaller, M.L., Evans, C.R., and Leiser, S.F. (2021). An alternative food source for metabolism and longevity studies in *Caenorhabditis elegans*. Commun Biol *4*, 258. 10.1038/s42003-021-01764-4.

4. Onken, B., and Driscoll, M. (2010). Metformin Induces a Dietary Restriction–Like State and the Oxidative Stress Response to Extend *C. elegans* Healthspan via AMPK, LKB1, and SKN-1. PLoS ONE *5*, e8758. 10.1371/journal.pone.0008758.

5. Cabreiro, F., Au, C., Leung, K.-Y., Vergara-Irigaray, N., Cochemé, H.M., Noori, T., Weinkove, D., Schuster, E., Greene, N.D.E., and Gems, D. (2013). Metformin retards aging in *C. elegans* by altering microbial folate and methionine metabolism. Cell *153*, 228-239. 10.1016/j.cell.2013.02.035.

6. Pryor, R., Norvaisas, P., Marinos, G., Best, L., Thingholm, L.B., Quintaneiro, L.M., De Haes, W., Esser, D., Waschina, S., Lujan, C., et al. (2019). Host-Microbe-Drug-Nutrient Screen Identifies Bacterial Effectors of Metformin Therapy. Cell *178*, 1299-1312 e1229. 10.1016/j.cell.2019.08.003.

7. Espada, L., Dakhovnik, A., Chaudhari, P., Martirosyan, A., Miek, L., Poliezhaieva, T., Schaub, Y., Nair, A., Doring, N., Rahnis, N., et al. (2020). Loss of metabolic plasticity underlies metformin toxicity in aged *Caenorhabditis elegans*. Nat Metab *2*, 1316-1331. 10.1038/s42255-020-00307-1.

8. Chen, J., Ou, Y., Li, Y., Hu, S., Shao, L.W., and Liu, Y. (2017). Metformin extends *C. elegans* lifespan through lysosomal pathway. *eLife 6*. 10.7554/*eLife*.31268.

9. Ma, T., Tian, X., Zhang, B., Li, M., Wang, Y., Yang, C., Wu, J., Wei, X., Qu, Q., Yu, Y., et al. (2022). Low-dose metformin targets the lysosomal AMPK pathway through PEN2. Nature *603*, 159-165. 10.1038/s41586-022-04431-8.

10. Wu, L., Zhou, B., Oshiro-Rapley, N., Li, M., Paulo, J.A., Webster, C.M., Mou, F., Kacergis, M.C., Talkowski, M.E., Carr, C.E., et al. (2016). An Ancient, Unified Mechanism for Metformin Growth Inhibition in *C. elegans* and Cancer. Cell *167*, 1705-1718 e1713. 10.1016/j.cell.2016.11.055.

11. Shi, X., Tarazona, P., Brock, T.J., Browse, J., Feussner, I., and Watts, J.L. (2016). A *Caenorhabditis elegans* model for ether lipid biosynthesis and function. J Lipid Res *57*, 265-275. 10.1194/jlr.M064808.

12. Devany, E., Zhang, X., Park, J.Y., Tian, B., and Kleiman, F.E. (2013). Positive and negative feedback loops in the p53 and mRNA 3' processing pathways. Proc Natl Acad Sci U S A *110*, 3351-3356. 10.1073/pnas.1212533110.

13. Perez-Diaz, L., Pastro, L., Smircich, P., Dallagiovanna, B., and Garat, B. (2013). Evidence for a negative feedback control mediated by the 3' untranslated region assuring the low expression level of the RNA binding protein TcRBP19 in T. cruzi epimastigotes. Biochem Biophys Res Commun *436*, 295-299. 10.1016/j.bbrc.2013.05.096.

14. Wu, L., Zhou, B., Oshiro-Rapley, N., Li, M., Paulo, J.A., Webster, C.M., Mou, F., Kacergis, M.C., Talkowski, M.E., Carr, C.E., et al. (2016). An Ancient, Unified Mechanism for Metformin Growth Inhibition in *C. elegans* and Cancer. Cell *167*, 1705-1718.e1713. 10.1016/j.cell.2016.11.055.

15. Stirling, D.R., Swain-Bowden, M.J., Lucas, A.M., Carpenter, A.E., Cimini, B.A., and Goodman, A. (2021). CellProfiler 4: improvements in speed, utility and usability. BMC Bioinformatics *22*, 433. 10.1186/s12859-021-04344-9.

16. Deng, J., Dai, Y., Tang, H., and Pang, S. (2020). SKN-1 Is a Negative Regulator of DAF-16 and Somatic Stress Resistance in *Caenorhabditis elegans*. G3 (Bethesda) *10*, 1707-1712. 10.1534/g3.120.401203.

17. Frankino, P.A., Siddiqi, T.F., Bolas, T., Bar-Ziv, R., Gildea, H.K., Zhang, H., Higuchi-Sanabria, R., and Dillin, A. (2022). SKN-1 regulates stress resistance downstream of amino catabolism pathways. iScience *25*, 104571. 10.1016/j.isci.2022.104571.

18. Steinbaugh, M.J., Narasimhan, S.D., Robida-Stubbs, S., Moronetti Mazzeo, L.E., Dreyfuss, J.M., Hourihan, J.M., Raghavan, P., Operana, T.N., Esmaillie, R., and Blackwell, T.K. (2015). Lipid-mediated regulation of SKN-1/Nrf in response to germ cell absence. *eLife 4*. 10.7554/*eLife*.07836.